# Continuous Submodular Maximization: Beyond DR-Submodularity

**Moran Feldman**
Department of Computer Science
University of Haifa
Haifa 3498838, Israel
moranfe@cs.haifa.ac.il

**Amin Karbasi**
School of Engineering and Applied Science
Yale University
New Haven, CT 06520
amin.karbasi@yale.edu

## Abstract

In this paper, we propose the first continuous optimization algorithms that achieve a constant factor approximation guarantee for the problem of monotone continuous submodular maximization subject to a linear constraint. We first prove that a simple variant of the vanilla coordinate ascent, called COORDINATE-ASCENT+, achieves a $(\frac{e-1}{2e-1} - \varepsilon)$-approximation guarantee while performing $O(n/\varepsilon)$ iterations, where the computational complexity of each iteration is roughly $O(n/\sqrt{\varepsilon} + n \log n)$ (here, $n$ denotes the dimension of the optimization problem). We then propose COORDINATE-ASCENT++, that achieves the tight $(1 - 1/e - \varepsilon)$-approximation guarantee while performing the same number of iterations, but at a higher computational complexity of roughly $O(n^3/\varepsilon^{2.5} + n^3 \log n/\varepsilon^2)$ per iteration. However, the computation of each round of COORDINATE-ASCENT++ can be easily parallelized so that the computational cost per machine scales as $O(n/\sqrt{\varepsilon} + n \log n)$.

## 1 Introduction

Submodularity is a fundamental concept in combinatorial optimization, usually associated with discrete set functions [Fujishige, 1991]. As submodular functions formalize the intuitive notion of diminishing returns, and thus provide a useful structure, they appear in a wide range of modern machine learning applications including various forms of data summarization [Lin and Bilmes, 2012, Mirzasoleiman et al., 2013], influence maximization [Kempe et al., 2003], sparse and deep representations [Balkanski et al., 2016, Oh Song et al., 2017], fairness Celis et al. [2016], Kazemi et al. [2018], experimental design [Harshaw et al., 2019], neural network interpretability Elenberg et al. [2017], human-brain mapping [Salehi et al., 2017], adversarial robustness [Lei et al., 2018], crowd teaching [Singla et al., 2014], to name a few. Moreover, submodularity ensures the tractability of the underlying combinatorial optimization problems as minimization of submodular functions can be done exactly and (constrained) maximization of submodular functions can be done approximately. For more details regarding the theory and applications of submodular functions in machine learning and signal processing, we refer the interested reader to the recent surveys by Buchbinder and Feldman [2018] and Tohidi et al. [2020].

To capture an even larger set of applications, while providing rigorous guarantees, the discrete notion of submodularity has been generalized in various directions, including adaptive and interactive submodualarity for sequential decision making problems [Golovin and Krause, 2011, Guillory and Bilmes, 2010], weak submodularity for general set functions with a bounded submodularity distance [Das and Kempe, 2011] and sequence submodularity for time series analysis [Tschiatschek et al., 2017, Mitrovic et al., 2019], among other variants.

Very recently, a surge of new applications in machine learning and statistics motivated researchers to study continuous submodular functions [Bach, 2015, Wolsey, 1982a], a large class of non-convex/non-

concave functions, which may be optimized efficiently. In particular, it has been shown that continuous submodular minimization can be done exactly Bach [2015]. In contrast, for continuous submodular maximization, it is usually assumed that the continuous function is not only submodular, but also has the extra condition of diminishing returns. Such functions are usually called continuous DR-submodular [Bian et al., 2017b]. We should highlight that even though in the discrete domain, submodularity and diminishing returns are equivalent; in the continuous domain, the diminishing returns condition implies continuous submodularity, but not vice versa.

In this paper, we propose the first algorithms that achieve constant factor approximation guarantees for the maximization of a monotone continuous submodular function subject to a linear constraint. More specifically, **our contributions** can be summarized as follows:

- We develop a variant of the coordinate ascent algorithm, called COORDINATE-ASCENT+, that achieves a $(\frac{e-1}{2e-1} - \varepsilon)$-approximation guarantee while performing $O(n/\epsilon)$ iterations, where the computational complexity of each iteration is $O(n\sqrt{B/\varepsilon} + n \log n)$. Here, $n$ and $B$ denote the dimension of the optimization problem and the $\ell_1$ radius of the constraint set, respectively.

- We then develop COORDINATE-ASCENT++, that achieves the tight $(1 - 1/e - \varepsilon)$ approximation guarantee while performing $O(n/\epsilon)$ iterations, where the computational complexity of each iteration is $O(n^3\sqrt{B}/\varepsilon^{2.5} + n^3 \log n/\varepsilon^2)$. Moreover, COORDINATE-ASCENT++ can be easily parallelized so that the computational complexity per machine in each round scales as $O(n\sqrt{B/\epsilon} + n \log n)$.

Notably, to establish these results, we do not assume that the continuous submodular function satisfies the diminishing returns condition.

## 1.1 Related Work

Continuous submodular functions naturally arise in many machine learning applications such as Adwords for e-commerce and advertising [Mehta et al., 2007, Devanur and Jain, 2012], influence and revenue maximization [Bian et al., 2017b], robust budget allocation [Staib and Jegelka, 2017], multi-resolution data summarization [Bian et al., 2017b], learning assignments [Golovin et al., 2014], experimental design [Chen et al., 2018b], and MAP inference for determinantal point processes [Gillenwater et al., 2012, Hassani et al., 2019]. Continuous submodular functions have also been studied in statistics as negative log-densities of probability distributions. These distributions are referred to as multivariate totally positive of order 2 (MTP2) [Fallat et al., 2017] and classical examples are the multivariate logistic, Gamma and $F$ distributions, as well as characteristic roots of random Wishart matrices [Karlin and Rinott, 1980].

The focus of the current work is to study continuous submodular maximization. Almost all the existing works in this area consider a proper subclass of continuous submodular functions, called continuous DR-submodular, which satisfy diminishing returns conditions. In particular, when first order information (i.e., exact or stochastic gradients) is available Hassani et al. [2017] showed that (stochastic) gradient ascent achieves $1/2$-approximation guarantee for monotone continuous DR-submodular functions subject to a general convex body constraint.[1] Interestingly, one can achieve the tight approximation guarantee of $1 - 1/e$ by using conditional gradient methods [Bian et al., 2017b] or its efficient stochastic variants [Mokhtari et al., 2018a, Karbasi et al., 2019, Zhang et al., 2020]. A simple variant of the conditional gradient methods can also be applied to non-monotone DR-submodular functions, which results in a $1/e$-approximation guarantee [Bian et al., 2017a, Mokhtari et al., 2018c, Hassani et al., 2019]. The only work, we are aware of, that goes beyond the above line of work, and considers also non-DR continuous submodular functions is a recent work by Niazadeh et al. [2018], which developed a polynomial time algorithm with a tight $1/2$-approximation guarantee for the problem of continuous submodular maximization subject to a box constraint.

Discrete and continuous submodular maximization problems are inherently related to one another through the multilinear extension [Calinescu et al., 2011a]. Indeed, maximization of the multilinear extension (along with a subsequent rounding) has led to the best theoretical results in many settings,

including submodular maximization subject to various complex constraints [Feldman et al., 2011, Chekuri et al., 2014, Buchbinder and Feldman, 2019], online and bandit submodular maximization [Zhang et al., 2019, Chen et al., 2018a], decentralized solution [Mokhtari et al., 2018b, Xie et al., 2019], and algorithms with low adaptivity complexity Chekuri and Quanrud [2019], Balkanski et al. [2019], Chen et al. [2019], Ene et al. [2019].

**Remark:** Soma and Yoshida [2018] considered the closely related problem of maximizing a non-DR submodular function subject to a cardinality constraint over the integer lattice. In principle, any algorithm for their setting can be applied to our setting using a reduction at the cost of a small decrease in the approximation guarantee and some increase in the time complexity.[2] However, an error was recently found in the analysis of the algorithm of [Soma and Yoshida, 2018], and therefore, this algorithm cannot currently be used to get an algorithm for our setting via the above mentioned reduction. It is likely that the algorithm of [Soma and Yoshida, 2018] can be fixed by introducing an enumeration step similar to the one we describe in Section 5, but such a modification will significantly increase the time complexity of their algorithm. In Appendix A in the supplementary material we give the reduction from our settings to the setting of [Soma and Yoshida, 2018] and also describe the error that was found in the algorithm of [Soma and Yoshida, 2018].

## 2 Preliminaries and Problem Formulation

We first recall a few standard definitions regarding submodular functions. Even though submodularity is mostly considered in the discrete domain, the notion can be naturally extended to arbitrary lattices [Fujishige, 1991]. To this end, let us consider a subset of $\mathbb{R}_+^d$ of the form $\mathcal{X} = \prod_{i=1}^d \mathcal{X}_i$ where each $\mathcal{X}_i$ is a compact subset of $\mathbb{R}_+$. A function $F \colon \mathcal{X} \to \mathbb{R}_+$ is *submodular* [Wolsey, 1982b] if for all $(\mathbf{x}, \mathbf{y}) \in \mathcal{X} \times \mathcal{X}$, we have

$$F(\mathbf{x}) + F(\mathbf{y}) \geq F(\mathbf{x} \vee \mathbf{y}) + F(\mathbf{x} \wedge \mathbf{y}) \ ,$$

where $\mathbf{x} \vee \mathbf{y} \doteq \max(\mathbf{x}, \mathbf{y})$ (component-wise) and $\mathbf{x} \wedge \mathbf{y} \doteq \min(\mathbf{x}, \mathbf{y})$ (component-wise). A submodular function is monotone if for any $\mathbf{x}, \mathbf{y} \in \mathcal{X}$ such that $\mathbf{x} \leq \mathbf{y}$, we have $F(\mathbf{x}) \leq F(\mathbf{y})$ (here, by $\mathbf{x} \leq \mathbf{y}$ we mean that every element of $\mathbf{x}$ is less than that of $\mathbf{y}$). The above definition includes the discrete notion of submodularity over a set by restricting each $\mathcal{X}_i$ to $\{0, 1\}$. In this paper, we mainly consider *continuous submodular* functions, where each $\mathcal{X}_i$ is a closed interval in $\mathbb{R}_+$. When $F$ is twice differentiable, a continuous function is submodular if and only if all cross-second-derivatives are non-positive [Bach, 2015], i.e.,

$$\forall i \neq j, \forall \mathbf{x} \in \mathcal{X}, \ \frac{\partial^2 F(\mathbf{x})}{\partial x_i \partial x_j} \leq 0 \ .$$

Thus, continuous submodular functions can be convex (e.g., $F(\mathbf{x}) = \sum_{i,j} \phi_{i,j}(x_i - x_j)$ for $\phi_{i,j}$ convex), concave (e.g., $F(\mathbf{x}) = g(\sum_{i=1}^n \lambda_i x_i)$ for $g$ concave and $\lambda_i$'s non-negative), and neither (e.g., quadratic program $F(\mathbf{x}) = \mathbf{x}^T H \mathbf{x}$ where all off-diagonal elements of $H$ are non-positive).

A proper subclass of continuous submodular functions are called *DR-submodular* [Bian et al., 2017b, Soma and Yoshida, 2015] if for all $\mathbf{x}, \mathbf{y} \in \mathcal{X}$ such that $\mathbf{x} \leq \mathbf{y}$, standard basis vector $\mathbf{e}_i \in \mathbb{R}_+^n$ and a non-negative number $z \in \mathbb{R}_+$ such that $z\mathbf{e}_i + \mathbf{x} \in \mathcal{X}$ and $z\mathbf{e}_i + \mathbf{y} \in \mathcal{X}$, it holds that $F(z\mathbf{e}_i + \mathbf{x}) - F(\mathbf{x}) \geq F(z\mathbf{e}_i + \mathbf{y}) - F(\mathbf{y})$.[3] One can easily verify that for a differentiable DR-submodular function the gradient is an antitone mapping, i.e., for all $\mathbf{x}, \mathbf{y} \in \mathcal{X}$ such that $\mathbf{x} \leq \mathbf{y}$ we have $\nabla F(\mathbf{x}) \geq \nabla F(\mathbf{y})$ [Bian et al., 2017b]. An important example of a DR-submodular function is the multilinear extension [Calinescu et al., 2011b]. Maybe the simplest example of a continuous submodular function that is not DR-submodular is the quadratic function $F(x) = x^T H x + h^T x + b$ where only the off-diagonal entries of $H$ are non-positive (and there is no restrictions on the diagonal entries). Moreover, as we mentioned in the introduction, continuous submodular functions naturally arise as the negative log-densities of probability distributions. For instance, a distribution $p$ on $\mathcal{X}$ is called MTP2 if $p(x)p(y) \leq p(x \vee y)p(x \wedge y)$ for all $x, y \in \mathcal{X} \subseteq \mathbb{R}$. MTP2 implies positive association between random variables. As an example, a multivariat Gaussian distibution is MTP2

if an only if its inverse covariance matrix has non-positive off-diagonal entries. Therefore, finding the the most likely configuration in this setting amounts to maximizing a continuous submodular function.

The definition of DR-submodular functions implies that decreasing coordinates of a vector affects the contribution of these coordinates to the value of the vector in at most a proportional way. More formally, if $\mathbf{x}$ and $\mathbf{y}$ are two vectors such that $\mathbf{x}, \mathbf{x} + \mathbf{y} \in \mathcal{X}$, then for every real value $c \in [0, 1]$ it holds that $F(\mathbf{x} + c\mathbf{y}) \geq F(\mathbf{x}) + c[F(\mathbf{x} + \mathbf{y}) - F(\mathbf{x})]$. This property is very helpful in algorithms designed for DR-submodular functions since it allows one to lower bound the increase in the value of a given solution $\mathbf{x}$ when some coordinate $i$ of $\mathbf{x}$ is increased even when the increase does not make $x_i$ as large as the value of coordinate $i$ in some optimal solution. For (non-DR) continuous submodular functions this cannot be done, and thus, a coordinate whose value cannot be increased to match its value in some optimal solution (due to the constraint and incorrect decisions already made by the algorithm) is basically useless for the algorithm; rendering the design of algorithms for (non-DR) continuous submodular functions much more challenging than for DR-submodular functions.

In this paper, we consider the following fundamental optimization problem

$$\max_{\mathbf{x} \in \mathcal{X}} F(\mathbf{x}) \text{ subject to } \|\mathbf{x}\|_1 \leq B, \tag{1}$$

where $F$ is a non-negative monotone continuous submodular function. Without loss of generality, we assume that each closed interval $\mathcal{X}_i$ is of the form $[0, u_i]$ since otherwise for $\mathcal{X}_i = [a_i, a_i + u_i]$ we can always define a corresponding continuous submodular function $G(\mathbf{x}) = F(\mathbf{x} + \mathbf{a})$, where $\mathbf{a} = [a_1, \ldots, a_n]$. Similarly, we assume w.l.o.g., that $u_i \leq B$ for every coordinate $i$. We also assume that $F$ is $L$-smooth, meaning that $\|\nabla F(\mathbf{x}) - \nabla F(\mathbf{y})\|_2 \leq L\|\mathbf{x} - \mathbf{y}\|_2$ for some $L \geq 0$ and for all $\mathbf{x}, \mathbf{y} \in \mathcal{X}$. Finally, note that replacing a linear constraint of the form $\sum_{i=1}^{n} w_i x_i \leq B$ (where $w_i > 0$) with $\|\mathbf{x}\|_1 \leq B$ does not change the nature of the problem. In this case, we can simply define a corresponding function $G(\mathbf{x}) = F\left(\sum_{i=1}^{n} x_i \mathbf{e}_i / w_i\right)$ and solve Problem (1). This change of course changes $L$ by a factor of $W = \min_{1 \leq i \leq n} w_i$. Prior to our work, no constant approximation guarantee was known for Problem (1).

## 3 Plain Coordinate Ascent

In this section we present our plain coordinate ascent algorithm and analyze its guarantee. Our algorithm uses as a black box an algorithm for a one dimensional optimization problem whose properties are summarized by the following proposition. We include the proof of this proposition in Appendix B in the supplementary material.

**Proposition 3.1.** *Given a point $\mathbf{x} \in [\mathbf{0}, \mathbf{u}]$, a coordinate $i \in [n]$, bounds $0 < a \leq b \leq u_i - x_i$ and a positive parameter $\varepsilon \in (0, 1)$ there is a polynomial time algorithm that runs in $O(\sqrt{B/\varepsilon} + \log(\varepsilon/a))$ time and returns a value $y \in [a, b]$ maximizing the ratio $F(\mathbf{x} + y\mathbf{e}_i)/y$ up to an additive error of $\varepsilon L$.*

Using the algorithm whose existence is guaranteed by the last proposition, we can now formally state our coordinate ascent algorithm as Algorithm 1. This algorithm gets a quality control parameter $\varepsilon \in (0, 1/4)$.

We begin the analysis of Algorithm 1 with the following observation that bounds its time complexity. The proof of this observation and some other proofs appearing later in this paper have been moved to Appendix D in the supplementary material due to space constraints. In a nutshell, the observation holds since each iteration of the main loop of Algorithm 1 producees progress in at least one of three ways: making $x_j = u_j$ for some coordinate $j$, making $\|\mathbf{x}\|_1$ as large as $B$, or increasing $\|\mathbf{x}\|_1$ by at least $\delta$.

**Observation 3.2.** *The main loop of Algorithm 1 makes at most $O(n/\varepsilon)$ iterations, each running in $O(n\sqrt{B/\varepsilon} + n\log n)$ time. Thus, the entire algorithm runs in $O(n^2\sqrt{B}/\varepsilon^{1.5} + n^2 \log n/\varepsilon)$ time.*

Fix now some feasible solution $y \in [\mathbf{0}, \mathbf{u}]$. Intuitively, we say that an iteration of the main loop of Algorithm 1 is *good* (with respect to $\mathbf{y}$) if, at the beginning of the iteration, the algorithm still has the option to increase each coordinate of $\mathbf{x}$ to be equal to the corresponding coordinate of $\mathbf{y}$, and this does not violate the constraint (if a coordinate of $\mathbf{x}$ is already larger than the corresponding coordinate of $\mathbf{y}$ that is fine as well). Formally, an iteration is good if the inequality $y_i - x_i \leq d_i'$ was true in this iteration for every coordinate $i \in C$ (before the vector $\mathbf{x}$ was updated at the end of the iteration). Let

**Algorithm 1:** COORDINATE-ASCENT $(\varepsilon)$

---

**1** Let $\mathbf{x} \leftarrow \mathbf{0}$ and $\delta \leftarrow \varepsilon B/n$.
**2 while** $\|\mathbf{x}\|_1 \leq B$ **do**
**3** $\quad$ Let $C \subseteq [n]$ be the set of coordinates $i \in [n]$ for which $x_i < u_i$ (i.e., these coordinates can be increased in $\mathbf{x}$ to some positive extent without violating feasibility).
**4** $\quad$ **for** *every* $i \in C$ **do**
**5** $\quad\quad$ Let $d_i'$ be the maximum amount by which $x_i$ can be increased without violating feasibility. Formally, $d_i' = \min\{u_i - x_i, B - \|\mathbf{x}\|_1\}$.
**6** $\quad\quad$ Use the algorithm suggested by Proposition 3.1 to find a value $d_i \in [\min\{d_i', \delta\}, d_i']$ maximizing $\frac{F(\mathbf{x}+d_i\mathbf{e}_i)-F(\mathbf{x})}{d_i}$ up to an additive error of $\varepsilon L$.
**7** $\quad$ Let $j$ be the coordinate of $C$ maximizing $\frac{F(\mathbf{x}+d_j\mathbf{e}_j)-F(\mathbf{x})}{d_j}$, and update $\mathbf{x} \leftarrow \mathbf{x} + d_j\mathbf{e}_j$.
**8 return** $\mathbf{x}$.

---

$\ell$ denote the number of good iterations of the main loop of Algorithm 1, and let us denote by $\mathbf{x}^{(h)}$ the value of $\mathbf{x}$ after $h$ iterations for every $0 \leq h \leq \ell$. Using this notation, we can now state and prove the following lemma, which provides a lower bound on the value of $\mathbf{x}$ after any number of (good) iterations of Algorithm 1.

**Lemma 3.3.** *For every vector* $\mathbf{y} \in [\mathbf{0}, \mathbf{u}]$ *and integer* $0 \leq h \leq \ell$, $F(\mathbf{x}^{(h)}) \geq (1 - e^{-\|\mathbf{x}^{(h)}\|_1/(\|y\|_1+\varepsilon B)}) \cdot F(\mathbf{y}) - \|\mathbf{x}^{(h)}\|_1 \cdot \varepsilon L.$

*Proof.* We prove the lemma by induction on $h$. For $h = 0$, $\|\mathbf{x}^{(h)}\|_1 = 0$, and the lemma follows from the non-negativity of $F$. Thus, it remains to prove the lemma for some $h > 0$ given that it holds for $h - 1$. From this point on we restrict our attention to iteration number $h$ of Algorithm 1, and thus, when we refer to variables such as $C$ and $d_i$, these variables should be understood as taking the values they are assigned in this iteration. Given this assumption, for every $i \in C$, let us now define a value $o_i$ that is closest to $y_i - x_i^{(h-1)}$ among all the values in the range to which $d_i$ can belong. Formally,

$$o_i = \min\{\max\{y_i - x_i^{(h-1)}, \min\{\delta, d_i'\}\}, d_i'\} = \max\{y_i - x_i^{(h-1)}, \min\{\delta, d_i'\}\} \quad \forall\, i \in C \ ,$$

where the equality holds since the fact that the iteration we consider is a good iteration implies $y_i - x_i^{(h-1)} \leq d_i'$. Since $o_i$ is a valid choice for $d_i$, we get by the definition of $d_i$ that

$$\frac{F(\mathbf{x}^{(h-1)} + d_i\mathbf{e}_i) - F(\mathbf{x}^{(h-1)})}{d_i} \geq \frac{F(\mathbf{x}^{(h-1)} + o_i\mathbf{e}_i) - F(\mathbf{x}^{(h-1)})}{o_i} - \varepsilon L \ .$$

Using the definition of $j$ and the submodularity of $F$, the last inequality implies

$$\frac{F(\mathbf{x}^{(h-1)} + d_j\mathbf{e}_j) - F(\mathbf{x}^{(h-1)})}{d_j} \geq \frac{\sum_{i\in C} o_i \cdot \frac{F(\mathbf{x}^{(h-1)}+d_i\mathbf{e}_i)-F(\mathbf{x}^{(h-1)})}{d_i}}{\sum_{i\in C} o_i} \quad\quad (2)$$

$$\geq \frac{\sum_{i\in C}[F(\mathbf{x}^{(h-1)} + o_i\mathbf{e}_i) - F(\mathbf{x}^{(h-1)})]}{\sum_{i\in C} o_i} - \varepsilon L \geq \frac{F(\mathbf{x}^{(h-1)} + \sum_{i\in C} o_i\mathbf{e}_i) - F(\mathbf{x}^{(h-1)})}{\sum_{i\in C} o_i} - \varepsilon L \ .$$

To understand the rightmost side of the last inequality, we need the following two bounds.

$$\sum_{i\in C} o_i \leq \sum_{i\in C} \max\{y_i, \delta\} \leq \sum_{i\in C} y_i + n\delta \leq \|\mathbf{y}\|_1 + \varepsilon B \ ,$$

and

$$\mathbf{x}^{(h-1)} + \sum_{i\in C} o_i\mathbf{e}_i \geq \mathbf{x}^{(h-1)} + \sum_{i\in C}(y_i - x_i^{(h-1)})\mathbf{e}_i \geq \mathbf{y} \ .$$

Plugging these bounds into Inequality (2), and using the monotonicity of $F$, we get

$$\frac{F(\mathbf{x}^{(h-1)} + d_j\mathbf{e}_j) - F(\mathbf{x}^{(h-1)})}{d_j} \geq \frac{F(\mathbf{y}) - F(\mathbf{x}^{(h-1)})}{\|\mathbf{y}\|_1 + \varepsilon B} - \varepsilon L \ .$$

Since $\mathbf{x}^{(h)} = \mathbf{x}^{(h-1)} + d_j \mathbf{e}_j$, the last inequality now yields the following lower bound on $F(\mathbf{x}^{(h)})$.

$$F(\mathbf{x}^{(h)}) \geq F(\mathbf{x}^{(h-1)}) + \frac{d_j}{\|\mathbf{y}\|_1 + \varepsilon B} \cdot [F(\mathbf{y}) - F(\mathbf{x}^{(h-1)})] - \varepsilon L d_j$$

$$\geq \left(1 - \frac{d_j}{\|\mathbf{y}\|_1 + \varepsilon B}\right) \cdot F(\mathbf{x}^{(h-1)}) + \frac{d_j}{\|\mathbf{y}\|_1 + \varepsilon B} \cdot F(\mathbf{y}) - \varepsilon L d_j \ .$$

Finally, plugging into the last inequality the lower bound on $F(\mathbf{x}^{(h-1)})$ given by the induction hypothesis, we get

$$F(\mathbf{x}^{(h)}) \geq \left(1 - \frac{d_j}{\|\mathbf{y}\|_1 + \varepsilon B}\right) \cdot \left\{(1 - e^{-\|\mathbf{x}^{(h-1)}\|_1/(\|\mathbf{y}\|_1 + \varepsilon B)}) \cdot F(\mathbf{y}) - \|\mathbf{x}^{(h-1)}\|_1 \cdot \varepsilon L\right\}$$

$$+ \frac{d_j}{\|\mathbf{y}\|_1 + \varepsilon B} \cdot F(\mathbf{y}) - \varepsilon L d_j$$

$$\geq \left(1 - e^{-(\|x^{(h-1)}\|_1 + d_j)/(\|\mathbf{y}\|_1 + \varepsilon B)}\right) \cdot F(\mathbf{y}) - (\|x^{(h-1)}\|_1 + d_j) \cdot \varepsilon L$$

$$= \left(1 - e^{-\|x^{(h)}\|_1/(\|\mathbf{y}\|_1 + \varepsilon B)}\right) \cdot F(\mathbf{y}) - \|x^{(h)}\|_1 \cdot \varepsilon L \ . \qquad \square$$

Our next objective is to get an approximation guarantee for Algorithm 1 based on the last lemma. Such a guarantee appears below as Corollary 3.5. However, to prove it we also need the following observation, which shows that lower bounding the value of $F(x)$ at some point during the execution of Algorithm 1 implies the same bound also for the value of the final solution of the algorithm.

**Observation 3.4.** *The value of $F(\mathbf{x})$ only increases during the execution of Algorithm 1.*

*Proof.* The observation follows from the monotonicity of $F$ since $d_j$ is always non-negative. $\square$

Let opt be some optimal solution vector.

**Corollary 3.5.** *Let $\mathbf{x}_{\mathsf{CA}}$ be the vector outputted by Algorithm 1, then $F(\mathbf{x}_{\mathsf{CA}}) \geq (1 - 1/e - B^{-1} \cdot \max_{i \in [n]} u_i - \varepsilon) \cdot F(\mathsf{opt}) - \varepsilon B L$.*

The guarantee of Corollary 3.5 is close to an approximation ratio of $1 - 1/e$ when the upper bound $u_i$ is small compared to $B$ for every $i \in [n]$. In the next two sections we describe enhanced versions of our coordinate ascent algorithm that give an approximation guarantee which is independent of this assumption. We note that, formally, the analyses of these enhanced versions are independent of Corollary 3.5. However, the machinery used to prove this corollary is reused in these analyses.

## 4  Fast Enhanced Coordinate Ascent

In this section we describe one simple and fast way to enhance the plain coordinate ascent algorithm from Section 3, leading to the algorithm that we name COORDINATE-ASCENT+. Before describing COORDINATE-ASCENT+ itself, let us give a different formulation for the guarantee of Algorithm 1.

**Lemma 4.1.** *There is a coordinate $j \in [n]$ such that the output $\mathbf{x}_{\mathsf{CA}}$ of Algorithm 1 has a value of at least $(1 - 1/e - 2\varepsilon) \cdot F(\mathsf{opt} - \mathsf{opt}_j \mathbf{e}_j) - \varepsilon B L$.*

*Proof.* In this proof we use the notation from Section 3, and consider the last iteration $\ell'$ during this execution in which there is no coordinate $i \in [n]$ such that $\mathsf{opt}_i - x_i > d_i'$ (where $x_i$ represents here its value at the beginning of the iteration). If $\ell'$ is the last iteration of Algorithm 1, then all the iterations of Algorithm 1 are good when we choose $\mathbf{y} = \mathsf{opt}$. Thus, for this choice of $\mathbf{y}$ we get $\|\mathbf{x}^{(\ell')}\|_1 = B$, and by Lemma 3.3 the value of the output $\mathbf{x}_{\mathsf{CA}} = \mathbf{x}^{(\ell')}$ of Algorithm 1 is at least

$$(1 - e^{-B/(\|\mathsf{opt}\|_1 + \varepsilon B)}) \cdot F(\mathsf{opt}) - \varepsilon B L \geq (1 - e^{-1} - \varepsilon) \cdot F(\mathsf{opt}) - \varepsilon B L \ ,$$

where the inequality holds since $\|\mathsf{opt}\|_1 \leq B$. This guarantee is stronger than the guarantee of the lemma (because of the monotonicity of $F$), and thus, completes the proof for the current case.

Consider now the case in which iteration $\ell'$ is not the last iteration of Algorithm 1. In this case we set $j$ to be some coordinate in $[n]$ for which the inequality $\mathsf{opt}_j - x_j^{(\ell')} > d_j'$ holds. Choosing

$\mathbf{y} = \mathsf{opt} - \mathsf{opt}_j \mathbf{e}_j$, we get that Algorithm 1 has at least $\ell'$ good iterations. There are now two cases to consider based on the relationship between $\|\mathbf{y}\|_1$ and $B$. If $\|\mathbf{y}\|_1 \geq B/2$, then Lemma 3.3 and Observation 3.4 imply together that the value of $\mathbf{x}_{\mathrm{CA}}$ is at least

$$F(\mathbf{x}_{\mathrm{CA}}) \geq F(\mathbf{x}^{(\ell')}) \geq (1 - e^{-(B-\mathsf{opt}_j)/(B+\varepsilon B - \mathsf{opt}_j)}) \cdot F(\mathsf{opt} - \mathsf{opt}_j \mathbf{e}_j) - \varepsilon BL$$
$$\geq (1 - e^{2\varepsilon - 1}) \cdot F(\mathsf{opt} - \mathsf{opt}_j \mathbf{e}_j) - \varepsilon BL \geq (1 - e^{-1} - 2\varepsilon) \cdot F(\mathsf{opt} - \mathsf{opt}_j \mathbf{e}_j) - \varepsilon BL \ ,$$

where the second inequality holds since $\|\mathbf{x}^{(\ell')}\|_1 \leq B$, but $\mathsf{opt}_j - x_j^{(\ell')} > d_j' = B - \|\mathbf{x}^{(\ell')}\|_1$ (the last equality holds since the inequalities $\mathsf{opt}_j - x_j^{(\ell')} > d_j'$ and $u_j \geq \mathsf{opt}_j$ exclude the possibility of $d_j' = u_j - x_j^{(\ell')}$). The penultimate inequality holds since, by our assumption, $B - \mathsf{opt}_j \geq \|\mathbf{y}\|_1 \geq B/2$.

It remains to consider the caes in which $\|\mathbf{y}\|_1 \leq B/2$. In this case, for every coordinate $i \in [n]$ we have $y_i - x_i \leq y_i \leq B/2 \leq B - \|\mathbf{x}\|_1$ as long as $\|\mathbf{x}\|_1 \leq B/2$. Thus, all the iterations of Algorithm 1 are good until $\|\mathbf{x}\|_1$ gets to a size lager than $B/2$; which implies $\|\mathbf{x}^{(\ell')}\|_1 \geq B/2$. Hence, Lemma 3.3 and Observation 3.4 allow us to lower bound $F(\mathbf{x}_{\mathrm{CA}})$ also by

$$F(\mathbf{x}_{\mathrm{CA}}) \geq F(\mathbf{x}^{(\ell')}) \geq (1 - e^{-(B/2)/(B/2+\varepsilon B)}) \cdot F(\mathsf{opt} - \mathsf{opt}_j \mathbf{e}_j) - \varepsilon BL$$
$$\geq (1 - e^{2\varepsilon - 1}) \cdot F(\mathsf{opt} - \mathsf{opt}_j \mathbf{e}_j) - \varepsilon BL \geq (1 - e^{-1} - 2\varepsilon) \cdot F(\mathsf{opt} - \mathsf{opt}_j \mathbf{e}_j) - \varepsilon BL \ ,$$

where the second inequality follows from the above discussion and the inequality $\|\mathbf{x}^{(\ell')}\|_1 \leq B$ which holds since $\mathbf{x}^{(\ell')}$ is a feasible solution. $\qquad\square$

We are now ready to present the enhanced algorithm COORDINATE-ASCENT+, which appears as Algorithm 2. The enhancement done in this algorithm, and its analysis, is related to an algorithm of Cohen and Katzir [2008] obtaining the same approximation guarantee for the special case of discrete monotone submodular functions.

---

**Algorithm 2:** COORDINATE-ASCENT+ $(\varepsilon)$

---

**1** Let $\mathbf{x}_{\mathrm{CA}}$ be the solution produced by Algorithm 1 when run with $\varepsilon$.
**2** Let $\mathbf{x}_{\mathrm{CA+}}$ be the best solution among the $n + 1$ solutions $\mathbf{x}_{\mathrm{CA}}$ and $\{u_i \cdot \mathbf{e}_i\}_{i \in [n]}$.
**3** **return** $\mathbf{x}_{\mathrm{CA+}}$.

---

It is clear that the time complexity of Algorithm 2 is dominated by the time complexity of Algorithm 1. Thus, we only need to analyze the approximation ratio of Algorithm 2. This is done by the next theorem, whose proofs relies on the fact that one of the solutions checked by Algorithm 2 is $u_j e_j$ for the coordinate $j$ whose existence is guaranteed by Lemma 4.1.

**Theorem 4.2.** *Algorithm 2 outputs a solution of value at least $\left(\frac{e-1}{2e-1} - 2\varepsilon\right) \cdot F(OPT) - \varepsilon BL \geq (0.387 - 2\varepsilon) \cdot F(OPT) - \varepsilon BL$. It has $O(n/\varepsilon)$ iterations, each running in $O(n\sqrt{B/\varepsilon} + n \log n)$ time, which yields a time complexity of $O(n^2 \sqrt{B}/\varepsilon^{1.5} + n^2 \log n/\varepsilon)$.*

## 5  Optimal Approximation Ratio

In this section we describe a more involved way to enhance the plain coordinate ascent algorithm from Section 3, which leads to the algorithm that we name COORDINATE-ASCENT++ and achieves the optimal approximation ratio of 1 - 1/e (up to some error term). This enhancement uses as a black box an algorithm for a one dimensional optimization problem whose properties are summarized by the following proposition. We include the proof of this proposition in Appendix C in the supplementary material.

**Proposition 5.1.** *Given a point $\mathbf{x} \in [\mathbf{0}, \mathbf{u}]$, a coordinate $i \in [n]$, a target value $F(\mathbf{x}) \leq v \leq F(\mathbf{x} \vee u_i \mathbf{e}_i)$ and a positive parameter $\varepsilon \in (0, 1)$, there is a polynomial time algorithm that runs in $O(\log(B/\varepsilon))$ time and returns a value $0 \leq y \leq u_i - x_i$ such that*

- $F(\mathbf{x} + y\mathbf{e}_i) \geq v - \varepsilon L.$

- *There is no value $0 \leq y' < y$ such that $F(\mathbf{x} + y'\mathbf{e}_i) \geq v$.*

We can now give a simplified version of COORDINATE-ASCENT++, which appears as Algorithm 3. For simplicity, we assume in the description and analysis of this algorithm that $n \geq 3$. If this is not the case, one can simulate it by adding dummy coordinates that do not affect the value of the objective function. Algorithm 3 starts by guessing two coordinates $h_1$ and $h_2$ that contribute a lot of value to opt. Then it constructs a solution $\mathbf{x}$ with a small support using two executions of the algorithm whose existence is guaranteed by Proposition 5.1, one execution for each one of the coordinates $h_1$ and $h_2$. It then completes the solution $\mathbf{x}$ into a full solution by executing Algorithm 1 after "contracting" the coordinates $h_1$ and $h_2$, i.e., modifying the objective function so that it implicitly assumes that these coordinates take the values they take in $\mathbf{x}$.

---

**Algorithm 3:** COORDINATE-ASCENT++ (SIMPLIFIED) $(\varepsilon)$

---

**1** Guess the coordinate $h_1 \in [n]$ maximizing $F(\mathsf{opt}_{h_1} \cdot \mathbf{e}_{h_1})$ and the coordinate $h_2 \in [n] \setminus \{h_1\}$ other than $h_1$ maximizing $F(\sum_{i \in \{h_1, h_2\}} \mathsf{opt}_i \cdot \mathbf{e}_i)$.

**2** Let $\mathbf{x} \leftarrow \mathbf{0}$.

**3** **for** $i = 1$ **to** $2$ **do**

**4**      Guess a value $v_i$ obeying

$$\max\{F(\mathbf{x}), F(\mathbf{x} + \mathsf{opt}_{h_i}\mathbf{e}_{h_i}) - \varepsilon \cdot F(\mathsf{opt})\} \leq v_i \leq F(\mathbf{x} + \mathsf{opt}_{h_i}\mathbf{e}_{h_i}) \ .$$

**5**      Let $y_i$ be the value returned by the algorithm whose existence is guaranteed by Proposition 5.1 given $\mathbf{x}$ as the input vector, the coordinate $h_i$ and the target value $v_i$.

**6**      Update $\mathbf{x} \leftarrow \mathbf{x} + y_i\mathbf{e}_{h_i}$.

**7** Execute Algorithm 1 on the instance obtained by removing the coordinates $h_1$ and $h_2$, replacing the objective function with $F'(\mathbf{x}') = F(\mathbf{x}' + \mathbf{x}) - F(\mathbf{x})$ and decreasing $B$ by $\|\mathbf{x}\|_1$. Let $\mathbf{x}_{\mathrm{CA}}$ be the output of Algorithm 1.

**8** Return $\mathbf{x} + \mathbf{x}_{\mathrm{CA}}$ (we denote this sum by $\mathbf{x}_{\mathrm{CA++}}$ in the analysis).

---

We begin the analysis of Algorithm 3 by bounding its time complexity.

**Observation 5.2.** *Assuming the guesses made by Algorithm 3 do not require any time, Algorithm 3 has $O(n/\varepsilon)$ iterations, each running in $O(n\sqrt{B/\varepsilon} + n \log n)$ time, which yields a time complexity of $O(n^2\sqrt{B}/\varepsilon^{1.5} + n^2 \log n/\varepsilon)$.*

The next step in the analysis of Algorithm 3 is proving some properties of the vector $\mathbf{x} = \sum_{i \in \{h_1, h_2\}} y_i \cdot \mathbf{e}_i$ produced by the first part of the algorithm. In a nutshell, these properties holds since the definition of $v_j$ and the properties of Proposition 5.1 show together that the value chosen for $x_{h_j}$ by the algorithm of Proposition 5.1 gives almost as much value as choosing $\mathsf{opt}_{h_j}$, but it never overestimates $\mathsf{opt}_{h_j}$.

**Lemma 5.3.** $F(\mathbf{x}) \geq F(\sum_{j=1}^{2} \mathsf{opt}_{h_j} \cdot \mathbf{e}_{h_j}) - 2\varepsilon \cdot F(\mathsf{opt}) - 2\varepsilon L$ *and* $\mathbf{x} \leq \sum_{j=1}^{2} \mathsf{opt}_{h_j} \cdot \mathbf{e}_{h_j}$.

We now ready to prove the approximation guarantee of Algorithm 3. Intuitively, this proof is based on simply adding up the lower bound on $F(\mathbf{x})$ given by Lemma 5.3 and the lower bound on $F'(\mathbf{x}_{\mathrm{CA}})$ given by Lemma 4.1. Some of the ideas used in the proof can be traced back to a recent result by Nutov [2020], who described an algorithm achieving $(1 - 1/e)$-approximation for the discrete version of the problem we consider (namely, maximizing a non-negative monotone discrete submodular function subject to a knapsack constraint) using $O(n^4)$ function evaluations.

**Lemma 5.4.** *Algorithm 3 outputs a vector $\mathbf{x}_{\mathrm{CA++}}$ whose value is at least $(1 - 1/e - 4\varepsilon) \cdot F(\mathsf{opt}) - \varepsilon(B + 2)L$.*

To get our final COORDINATE-ASCENT++ algorithm, we need to explain how to implement the guesses of Algorithm 3. The coordinates $h_1$ and $h_2$ can be guessed by simply iterating over all the possible pairs of two coordinates. Similarly, by the next observation, to get $v_i$ it suffices to try all the possible values in the set $\{F(\mathbf{x}) + \varepsilon j \cdot F(u_{h_i}\mathbf{e}_{h_i}) \mid j \text{ is a non-negative integer and } F(\mathbf{x}) + \varepsilon j \cdot F(u_{h_i}\mathbf{e}_{h_i}) \leq F(\mathbf{x} + u_{h_i}\mathbf{e}_{h_i})\}$. In the following, we refer to this set as $\mathcal{J}(\mathbf{x}, h_i)$.

**Observation 5.5.** *Consider the vector* $\mathbf{x}$ *at the point in which Algorithm 3 guesses the value* $v_i$*. Then, there exists a value in the set* $\mathcal{J}(\mathbf{x}, h_i)$ *obeying the requirements from* $v_i$*.*

Our final COORDINATE-ASCENT++ algorithm appears as Algorithm 4. By the above discussion, the number of iterations it makes exceeds the number of iterations given by Observation 5.2 only by a factor of

$$n^2 \cdot \prod_{i=1}^{2} |\mathcal{J}(\mathbf{x}, h_i)| \leq n^2 \cdot \prod_{i=1}^{2} \left(1 + \frac{F(\mathbf{x} + u_{h_i}\mathbf{e}_{h_i}) - F(\mathbf{x})}{\varepsilon \cdot F(u_{h_i}\mathbf{e}_{h_i})}\right) = O(\varepsilon^{-2}n^2) \ ,$$

where the equality holds since the submodulrity and non-negativity of $f$ imply $F(\mathbf{x} + u_{h_i}\mathbf{e}_{h_i}) - F(\mathbf{x}) \leq F(u_{h_i}\mathbf{e}_{h_i})$.

---

**Algorithm 4:** COORDINATE-ASCENT++ $(\varepsilon)$

---

**1** **for** *every pair of distinct coordinates* $h_1, h_2 \in [n]$ **do**
**2** $\quad$ Let $\mathbf{x}^{(0)} \leftarrow \mathbf{0}$.
**3** $\quad$ **for** *every* $v_1 \in \mathcal{J}(\mathbf{x}^{(0)}, h_1)$ **do**
**4** $\quad\quad$ Let $y_1$ be the value returned by the algorithm guaranteed by Proposition 5.1 given $\mathbf{x}^{(0)}$
$\quad\quad\quad$ as the input vector, the coordinate $h_1$ and the target value $v_1$.
**5** $\quad\quad$ Set $\mathbf{x}^{(1)} \leftarrow \mathbf{x}^{(0)} + y_1 \mathbf{e}_{h_1}$.
**6** $\quad\quad$ **for** *every* $v_2 \in \mathcal{J}(\mathbf{x}^{(1)}, h_2)$ **do**
**7** $\quad\quad\quad$ Let $y_2$ be the value returned by the algorithm guaranteed by Proposition 5.1 given
$\quad\quad\quad\quad$ $\mathbf{x}^{(1)}$ as the input vector, the coordinate $h_2$ and the target value $v_2$.
**8** $\quad\quad\quad$ Update $\mathbf{x} \leftarrow \mathbf{x}^{(1)} + y_2 \mathbf{e}_{h_2}$.
**9** $\quad\quad\quad$ Execute Algorithm 1 on the instance obtained by removing the coordinates $h_1$ and
$\quad\quad\quad\quad$ $h_2$, replacing the objective function with $F'(\mathbf{x}') = F(\mathbf{x}' + \mathbf{x}) - F(\mathbf{x})$ and
$\quad\quad\quad\quad$ decreasing $B$ by $\|\mathbf{x}\|_1$. Let $\mathbf{x}_{\text{CA}}$ be the output of Algorithm 1.
**10** $\quad\quad\quad$ Mark $\mathbf{x} + \mathbf{x}_{\text{CA}}$ as a candidate solution.

**11** Return the solution maximizing $F$ among all the solutions marked above as candidate
$\quad$ solutions.

---

The next theorem summarizes the result we have proved in this section.

**Theorem 5.6.** *For every* $\varepsilon \in (0, 1)$*, Algorithm 4 is an algorithm for our problem which produces a solution of value at least* $(1 - 1/e - 4\varepsilon) \cdot F(\text{opt}) - \varepsilon(B+2)L$*. It has* $O(n^3/\varepsilon^3)$ *iterations, each running in* $O(n\sqrt{B/\varepsilon} + n\log n)$ *time, which yields a time complexity of* $O(n^4\sqrt{B}/\varepsilon^{2.5} + n^4 \log n/\varepsilon^3)$*.*

In all the loops of Algorithm 4, the iterations are independent, and thus, can be done in parallel instead of sequentially. Thus, the parallel time required for Algorithm 4 is equal to the time complexity of Algorithm 3, which by Observation 5.2 is only $O(n^2\sqrt{B}/\varepsilon^{1.5} + n^2 \log n/\varepsilon)$.

## 6 Conclusion

In this paper, we provided the first constant factor approximation guarantees for the problem of maximizing a monotone continuous submodular function subject to a linear constraint. Crucially, our results did not rely on DR-submodularity.

A natural open problem is to extend our algorithms to more involved kinds constraints. To understand the difficult of such an extension, let us recall that our algorithms resemble algorithms for maximizing a submodular *set* function subject to a knapsack constraint even though all the coordinates have a coefficient of 1 in the constraint, which corresponds to a simpler cardinality constraint. Intuitively, this is the case because our algorithms treat monotone non-DR submodular continuous function as if they were monotone submodular *set* functions over an infinite ground set consisting of pairs $(i, v)$, where $i$ is a coordinate and $v$ is the value assigned to this coordinate. Unfortunately, this intuitive reduction from continuous to set functions converts cardinality constraints into knapsack constraints, forcing us to employ knapsack techniques even when handling cardinality-like constraints. Similarly, other kinds of constraints will also become more involved if the above intuitive reduction is applied to them, often leading to constraints that are difficult to handle.

## Broader Impact

In this paper, we provided the first constant factor approximation guarantees for a large class of non-convex functions with combinatorial structure. Since it is a theoretical result in nature, a broader impact discussion is not applicable.

## Acknowledgments and Disclosure of Funding

We would like to thank the anonymous referees for pointing out the work of Soma and Yoshida [2018] to us.

The work of Moran Feldman was supported in part by ISF grants no. 1357/16 and 459/20. Amin Karbasi is partially supported by NSF (IIS-1845032), ONR (N00014-19-1-2406), AFOSR (FA9550-18-1-0160), and TATA Sons Private Limited.

## Footnotes

[1]We note that the result of Hassani et al. [2017] can be viewed as an adaptation of an algorithm designed by Chekuri et al. [2011] for multilinear extensions of discrete monotone submodular functions.

[2]The reduction between the two settings is a bit subtle, and we would like to thank an anonymous referee for suggesting its crux idea to us.

[3]It is well-known that (non-DR) continuous submodularity is equivalent to requiring this inequality to hold whenever the vectors $\mathbf{x}$ and $\mathbf{y}$ also obey $x_i = y_i$.

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
