[Supplementary Material]

# A The Algorithm of [Soma and Yoshida, 2018]: Reduction and Error

An mentioned in Section 1, Soma and Yoshida [2018] described an algorithm for the problem of maximizing a monotone submodular function subject to a cardinality constraint over the integer lattice. Section A.1 gives a reduction (based on an idea suggested by an anonymous referee) from their problem to ours, that in principle can be used to convert any algorithm for their setting into an algorithm for our setting with a slightly worse approximation guarantee and a somewhat higher time complexity. However, an error was recently found in the analysis of the algorithm suggested by [Soma and Yoshida, 2018], and this error is described in Section A.2.

## A.1 Reduction from Our Problem to the Integer Lattice

The difference between our problem and the integral lattice setting of [Soma and Yoshida, 2018] is that in the last setting a solution is feasible only if all its coordinates are integral. Thus, an algorithm for the integer lattice setting can find a solution for our problem that approximates the best integral solution. Furthermore, using scaling it can also find a solution for our problem that approximates the best solution whose coordinates are all integer multiples of $\delta = B/(2n^3)$ (although this scaling might of course affect the time complexity of the algorithm). This means that an algorithm for the integer lattice that has some approximation guarantee provides roughly the same guarantee also when applied to our problem (after scaling) whenever the instance of our problem is nice in the following sense.

**Definition A.1.** *An instance of our problem is* nice *if the best solution for it whose coordinates are all integer multiples of $\delta$ is as good as the best solution for it without this restriction up to a multiplicative factor of $1 - O(n^{-1})$ and an additive factor of $O(n^{-1}) \cdot B^2 L$.*

Unfortunately, there are instances of our problem that are not nice. For example, consider an $n$ dimensional instance in which the objective function is simply $F(\mathbf{x}) = x_1$, and the upper bound vector $\mathbf{u}$ is

$$u_i = \begin{cases} \delta/2 & \text{if } i = 1 \ , \\ 1 & \text{otherwise} \ . \end{cases}$$

In this instance, every solution whose coordinates are all multiple integers of $\delta$ must have $0$ in the first coordinate, and thus, is of value $0$. Since the objective function is linear (and thus, $0$-smooth), the last observation implies that the above instance is nice only if the value of the optimal solution for it is also $0$, but this is clearly not the case.

To handle instances that are not nice, we need to make two observations. First, the vector $\mathbf{u}$ itself is an optimal solution whenever $\|\mathbf{u}\|_1 \leq B$ because of the monotonicity of $F$. Thus, we may assume from now on $\|\mathbf{u}\|_1 > B$. Second, the main reason that the instance in the above example failed to be nice is that the vector $\mathbf{u}$ in it included values that are not integer multiples of $\delta$. To solve this second issue, we consider an auxiliary instance defined as follows. Let $\mathbf{u}'$ be the vector obtained by rounding down the entries of $\mathbf{u}$ to the nearest integer multiples of $\delta$. The auxiliary instance has the domain $[0, \mathbf{u}']$, the budget $B' = B - \|\mathbf{u} - \mathbf{u}'\|_1$ and the objective function

$$F'(\mathbf{x}) = F(\mathbf{x} + \mathbf{u} - \mathbf{u}') \ .$$

**Lemma A.1.** *Assuming $\|\mathbf{u}\|_1 > B$, every feasible solution $\mathbf{x}'$ for the auxiliary instance can be converted into a solution $\mathbf{x}' + \mathbf{u} - \mathbf{u}'$ for the original instance that is also feasible and has the same value, and every feasible solution $\mathbf{x}$ for the original solution can be converted into a feasible solution $\mathbf{y}$ for the auxiliary solution whose coordinates are all integer multiples of $\delta = B/(2n^3)$ such that $F'(\mathbf{y}) \geq (1 - n^{-1}) \cdot F(\mathbf{x}) - B^2 L/(4n^3)$.*

*Proof.* To prove the first part of the lemma it suffices to observe that $F(\mathbf{x}' + \mathbf{u} - \mathbf{u}') = F'(\mathbf{x}')$ by the definition of $F'$ and $\|\mathbf{x}' + \mathbf{u} - \mathbf{u}'\|_1 = \|\mathbf{x}'\|_1 + \|\mathbf{u} - \mathbf{u}'\|_1 \leq B' + \|\mathbf{u} - \mathbf{u}'\|_1 = B$ because $\mathbf{x}'$ is feasible in the auxiliary instance. Thus, it remains to prove the second part of the lemma.

By the monotonicity of $F$ and the inequality $\|\mathbf{u}\| > B$, we may assume without loss of generality that $\|\mathbf{x}\|_1 = B$. Let us now denote by $i$ the maximal coordinate of $\mathbf{x}$, which implies $x_i \geq B/n$. We define the promised vector $\mathbf{y}$ as follows. For every coordinate $j \neq i$, $y_j$ is simply the smallest integer multiple of $\delta$ which is at least $x_j - (u_j - u'_j)$—note that $y_j$ is non-negative since $u_j - u'_j < \delta$ by the definition of $\mathbf{u}'$, and furthermore, $y_j$ is at most $u'_j$ since $u'_j$ is an integer multiple of $\delta$ and

$x_j - (u_j - u'_j) \leq u'_j$. For coordinate $i$, we define $y_i$ to be the largest multiple integer of $\delta$ which is at most $B' - \sum_{j \neq i} y_j$. One can observe that $y_i$ is also non-negative since

$$B' - \sum_{j \neq i} y_j \geq B' - \sum_{j \neq i} [x_j - (u_j - u'_j) + \delta]$$

$$\geq B' - \|x\|_1 + x_1 + (\|\mathbf{u}\|_1 - \|\mathbf{u}'\|_1) - n\delta = x_1 - n\delta \geq \frac{B}{n} - \frac{B}{2n^2} \geq \frac{B}{2n} \quad,$$

that $y_i$ is upper bounded by $u'_i$ since

$$B' - \sum_{j \neq i} y_j \leq B' - \sum_{j \neq i} [x_j - (u_j - u'_j)]$$

$$= B' - \|x\|_1 + (\|\mathbf{u}\|_1 - \|\mathbf{u}'\|_1) + x_i - u_i + u'_i = x_i - u_i + u'_i \leq u'_i \quad,$$

and that the definition of $y_i$ guarantees $\|\mathbf{y}\| \leq B'$. All these observations together imply that $y$ is indeed feasible in the auxiliary instance.

It remains to relate $F'(\mathbf{y})$ to $F(\mathbf{x})$. To do that, we need an hybrid vector $\mathbf{z}$ which is identical to $\mathbf{y}$ on all coordinates other than $i$, and is identical to $\mathbf{x} - \mathbf{u} + \mathbf{u}'$ in coordinate $i$. Clearly, $\mathbf{z} + \mathbf{u} - \mathbf{u}'$ dominates $\mathbf{x}$ coordinate-wise, and thus, by the monotonicity of $F$,

$$F'(\mathbf{z}) = F(\mathbf{z} + \mathbf{u} - \mathbf{u}') \geq F(\mathbf{x}) \quad.$$

We also note that

$$z_i - y_i = x_i - y_i = x_i - B' + \sum_{j \neq i} y_j \leq x_i - B' + \sum_{j \neq i} [x_j - (u_j - u'_j) + \delta]$$

$$\leq \|\mathbf{x}\|_1 - B' - (\|\mathbf{u}\|_1 - \|\mathbf{u}'\|_1) + n\delta = n\delta \quad.$$

Therefore, if we denote by $g$ the derivative of $F'$ at $\mathbf{y}$ according to the coordinate $i$, then the smoothness of $F$ implies

$$F'(\mathbf{z}) - F'(\mathbf{y}) = \int_0^{z_i - y_i} \frac{dF'}{dt}(\mathbf{y} + t\mathbf{e}_i)dt \leq \int_0^{z_i - y_i} (g + Lt)dt$$

$$= (z_i - y_i)g + \frac{L(z_i - y_i)^2}{2} \leq n\delta g + \frac{Ln^2\delta^2}{2} \leq \frac{2n^2\delta}{B} \cdot \left( \frac{Bg}{2n} - \frac{LB^2}{8n^2} \right) + \frac{\delta LB}{2}$$

$$= \frac{2n^2\delta}{B} \cdot \int_0^{B/(2n)} (g - Lt)dt + \frac{\delta LB}{2} \leq \frac{2n^2\delta}{B} \cdot \int_0^{B/(2n)} \frac{dF'}{dt}(\mathbf{y} - t\mathbf{e}_i)dt + \frac{\delta LB}{2}$$

$$\leq \frac{2n^2\delta}{B} \cdot F'(\mathbf{y}) + \frac{\delta LB}{2} = n^{-1} \cdot F'(\mathbf{y}) + \frac{LB^2}{4n^3} \quad.$$

Rearranging the last inequality, we finally get

$$F'(\mathbf{y}) \geq \frac{F'(\mathbf{z}) - LB^2/(4n^3)}{1 + n^{-1}}$$

$$\geq (1 - n^{-1}) \cdot F'(\mathbf{z}) - LB^2/(4n^3) \geq (1 - n^{-1}) \cdot F(\mathbf{x}) - LB^2/(4n^3) \quad. \qquad \square$$

Given the last lemma, it is clear that one can find an approximate optimal solution for the original instance by running an algorithm on the auxiliary instance for (approximately) finding the best feasible solution whose coordiantes are all integer multiples of $\delta$.

### A.2 The Error in the Algorithm of [Soma and Yoshida, 2018]

The algorithm of [Soma and Yoshida, 2018] involves a procedure named `BinarySearchLattice` whose job is to find a feasible augmentation for the current solution whose density passes some given threshold $\theta$ (or indicate that such an augmentation does not exist)—formally, Property (2) of Lemma 4 in [Soma and Yoshida, 2018] proves that the procedure `BinarySearchLattice` has this behavior. The main algorithm of [Soma and Yoshida, 2018] (which appears there as Algorithm 3) invokes `BinarySearchLattice` repeatably with a decreasing threshold $\theta$, and apply the augmentation found in every invocation (assuming one is found). The idea is to consider the set $A$ of augmentations

corresponding to increasing individual coordinates of the solution to their values in the optimal solution, with the assumption that, once the threshold $\theta$ becomes (roughly) equal to the density of some augmentation in $A$, the procedure `BinarySearchLattice` will find an augmentation of the same density to be applied.

If this assumption were correct, it would have led to an approximation guarantee because the average density of the augmentations in $A$ can be lower bounded. Unfortunately, however, for Lemma 4 to prove this assumption the augmentations of $A$ must all be feasible, which might not be the case once the $\ell_1$ norm of the solution becomes large enough. From a more formal point of view, the issue is that the proof of Lemma 5 of [Soma and Yoshida, 2018]—which is based on the idea described—invokes Lemma 4 with $k' + \Delta(a)$, but without verifying that $k' + \Delta(a)$ is upper bounded by $k_{\max}$ (which is a necessary condition of Lemma 4 equivalent to the intuitive notion that the augmentation represented by $k' + \Delta(a)$ should be feasible).

# B  Proof of Proposition 3.1

In this section we prove Proposition 3.1. Let us begin by restating this proposition.

**Proposition 3.1.** *Given a point $\mathbf{x} \in [\mathbf{0}, \mathbf{u}]$, a coordinate $i \in [n]$, bounds $0 < a \le b \le u_i - x_i$ and a positive parameter $\varepsilon \in (0, 1)$ there is a polynomial time algorithm that runs in $O(\sqrt{B/\varepsilon} + \log(\varepsilon/a))$ time and returns a value $y \in [a, b]$ maximizing the ratio $F(\mathbf{x} + y\mathbf{e}_i)/y$ up to an additive error of $\varepsilon L$.*

As might be expected, the algorithm we use to prove Proposition 3.1 tries a relatively small set of possible options for $y$, and then outputs the value yielding the maximum $F(\mathbf{x} + y\mathbf{e}_i)/y$ ratio. To define the set of values which the algorithm checks, we first need to define the following recursive series.

$$z_0 = a \qquad \text{and} \qquad z_i = z_{i-1} + \sqrt{\varepsilon z_{i-1}} \ .$$

Using this definition, we can now formally state the algorithm used to prove Proposition 3.1 as Algorithm 5.

---

**Algorithm 5:** ONE COORDINATE RATIO MAXIMIZER

---
**1** Let $M = \{b\} \cup \{z_i \mid i \text{ is a non-negative integer and } z_i \in [a, b]\}$.
**2** Return $y \in \arg\max_{y \in M} F(\mathbf{x} + y\mathbf{e}_i)/y$.

---

Before analyzing the quality of the solution returned by Algorithm 5, let us prove that it indeed has the required time complexity.

**Lemma B.1.** *The time complexity of Algorithm 5 is $O(\sqrt{B/\varepsilon} + \log(\varepsilon/a))$.*

*Proof.* Let $\ell$ be the smallest non-negative integer such that $z_\ell \ge B$. Clearly, the size of $M$ is upper bounded by $\ell + 3$ since $b \le u_i \le B$, and thus, the time complexity of Algorithm 5 is $O(\ell)$. Hence, to prove the lemma, it suffices to argue that there exists a positive integer $i' = O(\sqrt{B/\varepsilon} + \log(\varepsilon/a))$ such that $z_{i'} \ge B$, and thus, $O(\ell) = O(i') = O(\sqrt{B/\varepsilon} + \log(\varepsilon/a))$.

Observe that if $z_i \le \varepsilon$, then $z_{i+1} \ge 2z_i$. Thus, for $i_0 = \lceil \log_2(\varepsilon/a) \rceil$ we already get $z_{i_0} \ge \varepsilon$. Consider now the function $f(x) = \varepsilon(x^2 + 16)/16$. We would like to prove by induction that for every non-negative integer $i \ge 0$ we have $f(i) \le z_{i+i_0}$. For $i = 0$ this holds since $f(0) = \varepsilon \le z_{i_0}$. Assume now that this claim holds for some integer $i - 1 \ge 0$, and let us prove it for $i$. Since $z_{i-1+i_0} \ge f(i-1)$ by the induction hypothesis, it suffices to argue that $f(i) - f(i-1) \le z_{i+i_0} - z_{i-1+i_0} = \sqrt{\varepsilon z_{i-1+i_0}}$. By the definition of $f$,

$$
\begin{aligned}
f(i) - f(i-1) &= \frac{\varepsilon(i^2 + 16)}{16} - \frac{\varepsilon[(i-1)^2 + 16]}{16} = \frac{\varepsilon(2i - 1)}{16} \\
&\le \frac{\varepsilon \cdot 4\sqrt{(i-1)^2 + 16}}{16} = \sqrt{\varepsilon \cdot \frac{\varepsilon[(i-1)^2 + 16]}{16}} = \sqrt{\varepsilon \cdot f(i-1)} \le \sqrt{\varepsilon \cdot z_{i-1+i_0}} \ ,
\end{aligned}
$$

where the second inequality follows from the induction hypothesis, and the first inequality holds since for $i \geq 1$

$$2i - 1 \leq 4\sqrt{(i-1)^2 + 16} \iff (2i-1)^2 \leq 16[(i-1)^2 + 16]$$
$$\iff 4i^2 - 4i + 1 \leq 16i^2 - 32i + 272 \iff 0 \leq 12i^2 - 28i + 271 \; ,$$

and the last inequality holds for every $i$.

To complete the proof, it remains to observe that for $i' = \lceil \log_2(\varepsilon/a) \rceil + \lceil 4\sqrt{B/\varepsilon} \rceil = i_0 + \lceil 4\sqrt{B/\varepsilon} \rceil$ we have

$$z_{i'} \geq f(i' - i_0) = \frac{\varepsilon((i' - i_0)^2 + 16)}{16} \geq \frac{\varepsilon(16B/\varepsilon + 16)}{16} \geq B \; . \qquad \square$$

Our next objective is to show that the solution produced by Algorithm 5 approximately maximizes the ratio $F(\mathbf{x} + y\mathbf{e}_i)/y$ within the range $[a, b]$. Let $y^*$ be a value within this range that truly maximizes this ratio, and let $y_M$ be the largest value in the set $M$ which is not larger than $y^*$ (possibly $y^* = y_M$ if $y^* \in M$). We argue below that $F(\mathbf{x} + y^*\mathbf{e}_i)/y^* \leq F(\mathbf{x} + y_M\mathbf{e}_i)/y_M + \varepsilon L$, which completes the proof of Proposition 3.1 since the membership of $y_M$ in $M$ implies that the ratio $F(\mathbf{x} + y\mathbf{e}_i)/y$ for the value $y$ returned by Algorithm 5 is at least as good as $F(\mathbf{x} + y_M\mathbf{e}_i)/y_M$.

The next lemma gives us a simple upper bound on the ratio $F(\mathbf{x} + y^*\mathbf{e}_i)/y^*$.

**Lemma B.2.**
$$\frac{F(\mathbf{x} + y^*\mathbf{e}_i)}{y^*} \leq \frac{F(\mathbf{x} + y_M\mathbf{e}_i)}{y_M} + \frac{(y^* - y_M)^2 L}{2y_M} \; .$$

*Proof.* The derivative of $F(\mathbf{x} + y\mathbf{e}_i)/y$ by $y$ is

$$\frac{\frac{dF}{dy}(\mathbf{x} + y\mathbf{e}_i) \cdot y - F(\mathbf{x} + y\mathbf{e}_i)}{y^2} \; .$$

Since $y^*$ is a maximizer of this ratio, the above derivative must be zero in $y^*$, i.e., we get

$$\frac{dF}{dy}(\mathbf{x} + y^*\mathbf{e}_i) \cdot y^* - F(\mathbf{x} + y^*\mathbf{e}_i) = 0 \Rightarrow \frac{dF}{dy}(\mathbf{x} + y^*\mathbf{e}_i) = \frac{F(\mathbf{x} + y^*\mathbf{e}_i)}{y^*} \; .$$

This allows us to use the smoothness of $F$ to upper bound the difference between $F(\mathbf{x} + y^*\mathbf{e}_i)$ and $F(\mathbf{x} + y_M\mathbf{e}_i)$ by

$$F(\mathbf{x} + y^*\mathbf{e}_i) - F(\mathbf{x} + y_M\mathbf{e}_i) = \int_{y_M}^{y^*} \frac{dF}{dy}(\mathbf{x} + y\mathbf{e}_i) dy \leq \int_{y_M}^{y^*} \left[ \frac{dF}{dy}(\mathbf{x} + y^*\mathbf{e}_i) + (y^* - y)L \right] dy$$

$$= \int_{y_M}^{y^*} \left[ \frac{F(\mathbf{x} + y^*\mathbf{e}_i)}{y^*} + (y^* - y)L \right] dy = \frac{y^* - y_M}{y^*} \cdot F(\mathbf{x} + y^*\mathbf{e}_i) + \frac{(y^* - y_M)^2 L}{2} \; .$$

Rearranging the last inequality, we get

$$\frac{y_M}{y^*} \cdot F(\mathbf{x} + y^*\mathbf{e}_i) \leq F(\mathbf{x} + y_M\mathbf{e}_i) + \frac{(y^* - y_M)^2 L}{2} \; ,$$

and the observation follows by dividing the last inequality by $y_M$. $\qquad \square$

Given the above discussion, the last lemma implies that to prove Proposition 3.1 we only need to argue that $\frac{(y^* - y_M)^2 L}{2y_M}$ is always upper bounded by $\varepsilon L$. The following observation shows that this is indeed the case.

**Observation B.3.** $\frac{(y^* - y_M)^2}{2y_M} \leq \varepsilon$.

*Proof.* By the definition of the set $M$, the value of $y^*$ must be at most $y_M + \sqrt{\varepsilon y_M}$. Thus,

$$\frac{(y^* - y_M)^2}{2y_M} \leq \frac{(\sqrt{\varepsilon y_M})^2}{2y_M} = \frac{\varepsilon y_M}{2y_M} = \frac{\varepsilon}{2} < \varepsilon \; . \qquad \square$$

# C   Proof of Proposition 5.1

In this section we prove Proposition 5.1. Let us begin by restating this proposition.

**Proposition 5.1.** *Given a point $\mathbf{x} \in [\mathbf{0}, \mathbf{u}]$, a coordinate $i \in [n]$, a target value $F(\mathbf{x}) \leq v \leq F(\mathbf{x} \vee u_i \mathbf{e}_i)$ and a positive parameter $\varepsilon \in (0, 1)$, there is a polynomial time algorithm that runs in $O(\log(B/\varepsilon))$ time and returns a value $0 \leq y \leq u_i - x_i$ such that*

- *$F(\mathbf{x} + y\mathbf{e}_i) \geq v - \varepsilon L$.*

- *There is no value $0 \leq y' < y$ such that $F(\mathbf{x} + y'\mathbf{e}_i) \geq v$.*

The algorithm we use to prove Proposition 5.1 has two phases. In the first phase, the algorithm uses binary search to zoom in on a small range of $y$ values which includes the lowest $y$ value for which $F(\mathbf{x} + y\mathbf{e}_i) = v$. Then, in the second phase, the algorithm uses linear interpolation to pick a value $y$ from this range for which $F(\mathbf{x} + y\mathbf{e}_i)$ is close to $v$. The linear interpolation parameters have to be selected with care to make sure that the value picked obeys the second guarantee of the proposition. A formal statement of the algorithm appears as Algorithm 6.

---

**Algorithm 6:** ONE COORDINATE GETTING TARGET VALUE

1 Let $a = 0$ and $b = u_i - x_i$.
2 **while** $b - a \geq \varepsilon$ **do**
3      Let $m = (b - a)/2$.
4      **if** $F(\mathbf{x} + m\mathbf{e}_i) \geq v$ **then** Update $b \leftarrow m$.
5      **else** Update $a \leftarrow m$.
6 Let $d \leftarrow \frac{F(\mathbf{x}+b\mathbf{e}_i) - F(\mathbf{x}+a\mathbf{e}_i)}{b-a} + \frac{\varepsilon L}{2}$, and $r \leftarrow \frac{v - F(\mathbf{x}+a\mathbf{e}_i)}{d}$.
7 Return $a + r$.

---

We begin the analysis of Algorithm 6 by showing that has the time complexity guaranteed by Proposition 5.1.

**Observation C.1.** *The time complexity of Algorithm 6 is at most $O(\log(B/\varepsilon))$.*

*Proof.* The time complexity of Algorithm 6 is proportional to the number of iterations made by the binary search in the first phase of the algorithm. Since this binary search starts with a range of size $u_i - x_i \leq u_i \leq B$, and ends when its range shrinks to a size of $\varepsilon$ or less, the number of iterations it performs is upper bounded by $\lceil \log(B/\varepsilon) \rceil$. $\qquad\square$

Let us denote now by $a_0$ and $b_0$ the values of the variables $a$ and $b$ when the binary search phase of Algorithm 6 terminates. By the design of the binary search, it is clear that $F(\mathbf{x} + a_0 \mathbf{e}_i) \leq v$. Furthermore, this inequality can hold as an equality only when $a_0 = 0$. Let us now get bounds on the derivative of $F(\mathbf{x} + y\mathbf{e}_i)$ as a function of $y$ within the range $[a_0, b_0]$.

**Lemma C.2.** *For every $y' \in [a_0, b_0]$, $\frac{dF}{dy}(\mathbf{x} + y\mathbf{e}_i) \in [d - \varepsilon L, d]$.*

*Proof.* By the smoothness of the function $F$

$$F(\mathbf{x} + b_0\mathbf{e}_i) - F(\mathbf{x} + a_0\mathbf{e}_i) = \int_{a_0}^{b_0} \frac{dF}{dy}(\mathbf{x} + y\mathbf{e}_i)dy \leq \int_{a_0}^{b_0} \left[ \frac{dF}{dy}(\mathbf{x} + y'\mathbf{e}_i) + |y - y'|L \right] dy$$

$$\leq (b_0 - a_0) \cdot \frac{dF}{dy}(\mathbf{x} + y'\mathbf{e}_i) + \frac{(b_0 - a_0)^2 L}{2}$$

$$\leq (b_0 - a_0) \cdot \frac{dF}{dy}(\mathbf{x} + y'\mathbf{e}_i) + \frac{(b_0 - a_0)\varepsilon L}{2} \ .$$

Dividing the last inequality by $b_0 - a_0$, we get

$$d - \frac{\varepsilon L}{2} \leq \frac{dF}{dy}(\mathbf{x} + y'\mathbf{e}_i) + \frac{\varepsilon L}{2} \implies d - \varepsilon L \leq \frac{dF}{dy}(\mathbf{x} + y'\mathbf{e}_i) \ .$$

Similarly, the smoothness of $F$ also implies

$$F(\mathbf{x} + b_0\mathbf{e}_i) - F(\mathbf{x} + a_0\mathbf{e}_i) = \int_{a_0}^{b_0} \frac{dF}{dy}(\mathbf{x} + y\mathbf{e}_i)dy \geq \int_{a_0}^{b_0} \left[\frac{dF}{dy}(\mathbf{x} + y'\mathbf{e}_i) - |y - y'|L\right] dy$$

$$\geq (b_0 - a_0) \cdot \frac{dF}{dy}(\mathbf{x} + y'\mathbf{e}_i) - \frac{(b_0 - a_0)^2 L}{2}$$

$$\geq (b_0 - a_0) \cdot \frac{dF}{dy}(\mathbf{x} + y'\mathbf{e}_i) - \frac{(b_0 - a_0)\varepsilon L}{2} \quad,$$

and this time dividing the last inequality by $b_0 - a_0$ yields

$$d - \frac{\varepsilon L}{2} \geq \frac{dF}{dy}(\mathbf{x} + y'\mathbf{e}_i) - \frac{\varepsilon L}{2} \implies d \geq \frac{dF}{dy}(\mathbf{x} + y'\mathbf{e}_i) \ . \qquad \square$$

The following corollary now completes the proof of Proposition 5.1 since the output of Algorithm 6 is $a_0 + r$.

**Corollary C.3.** $F(\mathbf{x} + (a_0 + r)\mathbf{e}_i) \geq v - \varepsilon L$, and furthermore, $F(\mathbf{x} + y'\mathbf{e}_i) < v$ for every $0 \leq y' < a_0 + r$.

*Proof.* Using Lemma C.2, we get

$$F(\mathbf{x} + (a_0 + r)\mathbf{e}_i) = F(\mathbf{x} + a_0\mathbf{e}_i) + \int_{a_0}^{a_0+r} \frac{dF}{dy}(\mathbf{x} + y\mathbf{e}_i)dy$$

$$\geq F(\mathbf{x} + a_0\mathbf{e}_i) + \int_{a_0}^{a_0+r} (d - \varepsilon L)dy = F(\mathbf{x} + a_0\mathbf{e}_i) + r(d - \varepsilon L)$$

$$= F(\mathbf{x} + a_0\mathbf{e}_i) + [v - F(\mathbf{x} + a_0\mathbf{e}_i)] - r\varepsilon L$$

$$\geq F(\mathbf{x} + a_0\mathbf{e}_i) + [v - F(\mathbf{x} + a_0\mathbf{e}_i)] - \varepsilon L \quad,$$

where the third equality holds by plugging in the definition of $r$, and the last inequality holds since

$$r = \frac{v - F(\mathbf{x} + a\mathbf{e}_i)}{d} \leq \frac{F(\mathbf{x} + b\mathbf{e}_i) - F(\mathbf{x} + a\mathbf{e}_i)}{d} \leq b - a \leq \varepsilon < 1 \ .$$

Similarly, Lemma C.2 also implies for every $a_0 \leq y' < a_0 + r$

$$F(\mathbf{x} + y'\mathbf{e}_i) = F(\mathbf{x} + a_0\mathbf{e}_i) + \int_{a_0}^{y'} \frac{dF}{dy}(\mathbf{x} + y\mathbf{e}_i)dy$$

$$\leq F(\mathbf{x} + a_0\mathbf{e}_i) + \int_{a_0}^{y'} d\, dy = F(\mathbf{x} + a_0\mathbf{e}_i) + (y' - a_0)d$$

$$< F(\mathbf{x} + a_0\mathbf{e}_i) + rd = F(\mathbf{x} + a_0\mathbf{e}_i) + [v - F(\mathbf{x} + a_0\mathbf{e}_i)] = v \ .$$

If $r > 0$, then the last inequality completes the proof of the corollary because the monotonicity of $F$ guarantees $F(\mathbf{x} + y'\mathbf{e}_i) \leq F(\mathbf{x} + a_0\mathbf{e}_i) < v$ for every $0 \leq y' < a_0$. Thus, it remains to consider the case of $r = 0$. This case happens only when $F(\mathbf{x} + a_0\mathbf{e}_i) = v$, which implies by the discussion before Lemma C.2 that $a_0 = 0$ as well. Hence, the requirement $F(\mathbf{x} + y'\mathbf{e}_i) < v$ for every $0 \leq y' < a_0 + r$ is trivial in this case. $\qquad \square$

## D Missing Proofs

In this section we give the formal proofs which have been omitted from the main part of this paper.

### D.1 Proof of Observation 3.2

**Observation 3.2.** *The main loop of Algorithm 1 makes at most $O(n/\varepsilon)$ iterations, each running in $O(n\sqrt{B/\varepsilon} + n\log n)$ time. Thus, the entire algorithm runs in $O(n^2\sqrt{B}/\varepsilon^{1.5} + n^2 \log n/\varepsilon)$ time.*

*Proof.* We note that the way in which the algorithm assigns a value to $d_j$ implies that in any iteration of the main loop of Algorithm 1 one of the following must happen.

1. One option is that $d_j = u_j - x_j$. When this happens, the value of $x_j$ becomes equal to $u_j$, and thus, this is the last iteration in which the coordinate $j$ belongs to the set $C$.

2. Another option is that $d_j = B - \|\mathbf{x}\|_1$. In this case, $\|\mathbf{x}\|_1$ becomes equal to $B$ following the iteration, and thus, the algorithm terminates following this iteration.

3. If neither of the previous options happens, then the value $\|\mathbf{x}\|_1$ increases by at least $\delta$ following the iteration.

There can be at most $n$ iterations in which Option 1 happens since there are only $n$ coordinates, at most a single iteration in which Option 2 happens and at most $B/\delta = n/\varepsilon$ iterations in which Option 3 happens (since the value of $\|\mathbf{x}\|_1$ cannot exceed $B$). Thus, the total number of iterations is at most

$$n + 1 + \frac{n}{\varepsilon} = O(\varepsilon^{-1}n) \ .$$

We now note that every single iteration of the main loop of Algorithm 1 requires $O(n)$ time plus the time required for up to $n$ executions of the algorithm whose existence is guaranteed by Proposition 3.1. Furthermore, we can assume that each execution of the last algorithm gets $a = \delta$ because we always look for $d_i$ either inside a range containing a single value or a range whose lower bound is $\delta$. Thus, the time required for each such execution is upper bounded by

$$O\left(\sqrt{\frac{B}{\varepsilon}} + \log\left(\frac{\varepsilon}{\delta}\right)\right) = O\left(\sqrt{\frac{B}{\varepsilon}} + \log\left(\frac{n}{B}\right)\right) = O(\sqrt{B/\varepsilon} + \log n) \ ,$$

and the space required for the entire iteration of the main loop of Algorithm 1 is at most

$$n \cdot O(\sqrt{B/\varepsilon} + \log n) + O(n) = O(n\sqrt{B/\varepsilon} + n\log n) \ . \qquad \square$$

### D.2 Proof of Corollary 3.5

**Corollary 3.5.** *Let $\mathbf{x}_{\mathrm{CA}}$ be the vector outputted by Algorithm 1, then $F(\mathbf{x}_{\mathrm{CA}}) \geq (1 - 1/e - B^{-1} \cdot \max_{i \in [n]} u_i - \varepsilon) \cdot F(\mathsf{opt}) - \varepsilon BL$.*

*Proof.* By Observation 3.4, it suffices to argue that

$$F(\mathbf{x}^{(\ell)}) \geq (1 - 1/e - B^{-1} \cdot \max_{i \in [n]} u_i - \varepsilon) \cdot F(\mathsf{opt}) - \varepsilon BL \ .$$

Thus, in the rest of the proof we prove this inequality.

Plugging $\mathbf{y} = \mathsf{opt}$ into Lemma 3.3, we get

$$F(\mathbf{x}^{(\ell)}) \geq (1 - e^{-\|\mathbf{x}^{(\ell)}\|_1/(\|\mathbf{y}\|_1 + \varepsilon B)}) \cdot F(\mathsf{opt}) - \|\mathbf{x}^{(\ell)}\|_1 \cdot \varepsilon L \qquad (3)$$

$$\geq (1 - e^{-(1-\varepsilon)\|\mathbf{x}^{(\ell)}\|_1/B}) \cdot F(\mathsf{opt}) - \varepsilon BL \ ,$$

where the second inequality holds since $\|\mathbf{y}\|_1$ and $\|\mathbf{x}^{(\ell)}\|_1$ are both upper bounded by $B$. If iteration number $\ell$ is not the last iteration of Algorithm 1, then the fact that iteration number $\ell + 1$ was not a good iteration implies the existence of a coordinate $i \in [n]$ such that $y_i - x_i^{(\ell)} > d_i' = B - \|\mathbf{x}^{(\ell)}\|_1$ (the last equality holds since the inequalities $y_i - x_i^{(\ell)} > d_j'$ and $u_i \geq y_i$ exclude the possibility of $d_i' = u_i - x_i^{(\ell)}$). Thus, we get in this case

$$\|\mathbf{x}^{(\ell)}\|_1 > B - y_i + x_i^{(\ell)} \geq B - u_i \geq B - \max_{i \in [n]} u_i \ .$$

Moreover, the last inequality holds also in the case in which iteration number $\ell$ is the last iteration of Algorithm 1 because in this case $\|\mathbf{x}^{(\ell)}\|_1 = B$. Plugging this into Inequality (3), we get

$$F(\mathbf{x}^{(\ell)}) \geq (1 - e^{(1-\varepsilon)(\max_{i \in [n]} u_i/B - 1)}) \cdot F(\mathsf{opt}) - \varepsilon BL$$

$$\geq (1 - e^{\varepsilon + \max_{i \in [n]} u_i/B - 1}) \cdot F(\mathsf{opt}) - \varepsilon BL$$

$$\geq (1 - e^{-1} - B^{-1} \cdot \max_{i \in [n]} u_i - \varepsilon) \cdot F(\mathsf{opt}) - \varepsilon BL \ ,$$

where the last inequality holds since $e^{x-1} \leq e^{-1} + x$ for $x \in [0, 1.5]$. $\qquad \square$

## D.3 Proof of Theorem 4.2

**Theorem 4.2.** *Algorithm 2 outputs a solution of value at least $\left(\frac{e-1}{2e-1} - 2\varepsilon\right) \cdot F(OPT) - \varepsilon BL \geq (0.387 - 2\varepsilon) \cdot F(OPT) - \varepsilon BL$. It has $O(n/\varepsilon)$ iterations, each running in $O(n\sqrt{B/\varepsilon} + n\log n)$ time, which yields a time complexity of $O(n^2\sqrt{B}/\varepsilon^{1.5} + n^2\log n/\varepsilon)$.*

*Proof.* By Lemma 4.1, there exists a coordinate $j \in [n]$ such that

$$F(g) \geq (1 - e^{-1} - 2\varepsilon) \cdot F(\mathsf{opt} - \mathsf{opt}_j \mathbf{e}_j) - \varepsilon BL \ .$$

Since Algorithm 2 picks a solution $\mathbf{x}_{\mathrm{CA+}}$ that is at least as good as both $\mathbf{x}_{\mathrm{CA}}$ and $u_j\mathbf{e}_j$, the last inequality and the monotonicity of $F$ imply together that

$$\begin{aligned}
F(\mathbf{x}_{\mathrm{CA+}}) &\geq \frac{1}{2 - e^{-1} - 2\varepsilon} \cdot F(g) + \frac{1 - e^{-1} - 2\varepsilon}{2 - e^{-1} - 2\varepsilon} \cdot F(u_j\mathbf{e}_j) \\
&\geq \frac{1}{2 - e^{-1} - 2\varepsilon} \cdot \left[(1 - e^{-1} - 2\varepsilon) \cdot F(\mathsf{opt} - \mathsf{opt}_j\mathbf{e}_j) - \varepsilon BL\right] + \frac{1 - e^{-1} - 2\varepsilon}{2 - e^{-1} - 2\varepsilon} \cdot F(\mathsf{opt}_j\mathbf{e}_j) \\
&\geq \frac{1 - e^{-1} - 2\varepsilon}{2 - e^{-1} - 2\varepsilon} \cdot \left[F(\mathsf{opt} - \mathsf{opt}_j\mathbf{e}_j) + F(\mathsf{opt}_j\mathbf{e}_j)\right] - \varepsilon BL \\
&\geq \frac{1 - e^{-1} - 2\varepsilon}{2 - e^{-1} - 2\varepsilon} \cdot F(\mathsf{opt}) - \varepsilon BL \geq \left(\frac{1 - e^{-1}}{2 - e^{-1}} - 2\varepsilon\right) \cdot F(\mathsf{opt}) - \varepsilon BL \\
&= \left(\frac{e - 1}{2e - 1} - 2\varepsilon\right) \cdot F(\mathsf{opt}) - \varepsilon BL \ ,
\end{aligned}$$

where the penultimate inequality holds by the submodularity of $F$. $\qquad\square$

## D.4 Proof of Observation 5.2

**Observation 5.2.** *Assuming the guesses made by Algorithm 3 do not require any time, Algorithm 3 has $O(n/\varepsilon)$ iterations, each running in $O(n\sqrt{B/\varepsilon} + n\log n)$ time, which yields a time complexity of $O(n^2\sqrt{B}/\varepsilon^{1.5} + n^2\log n/\varepsilon)$.*

*Proof.* Besides the two executions of the algorithm whose existence is guaranteed by Proposition 5.1 and the execution of Algorithm 1, Algorithm 3 uses only constant time. Thus, the time complexity of Algorithm 3 is upper bounded by the sum of the time complexities of the two other algorithms mentioned. Furthermore, by Proposition 5.1, the total time complexity of the algorithm whose existence is guaranteed by this proposition is only

$$O\left(\log\left(\frac{B}{\varepsilon}\right)\right) \ ,$$

which is upper bounded by the time complexity of a single iteration of Algorithm 1 as given by Observation 3.2. Hence, both the number of iterations and the time per iteration of Algorithm 3 are asymptotically identical to the corresponding values for Algorithm 1. $\qquad\square$

## D.5 Proof of Lemma 5.3

**Lemma 5.3.** *$F(\mathbf{x}) \geq F(\sum_{j=1}^{2} \mathsf{opt}_{h_j} \cdot \mathbf{e}_{h_j}) - 2\varepsilon \cdot F(\mathsf{opt}) - 2\varepsilon L$ and $\mathbf{x} \leq \sum_{j=1}^{2} \mathsf{opt}_{h_j} \cdot \mathbf{e}_{h_j}$.*

*Proof.* Recall that the support of $\mathbf{x}$ contains only the coordinates $h_1$ and $h_2$. Thus, to prove the lemma, it suffices to argue that for every $i \in \{1, 2\}$

$$F\left(\sum_{j=1}^{i} x_{h_j} \cdot \mathbf{e}_{h_j}\right) - F\left(\sum_{j=1}^{i-1} x_{h_j} \cdot \mathbf{e}_{h_j}\right) \tag{4}$$

$$\geq F\left(\sum_{j=1}^{i} \mathsf{opt}_{h_j} \cdot \mathbf{e}_{h_j}\right) - F\left(\sum_{j=1}^{i-1} \mathsf{opt}_{h_j} \cdot \mathbf{e}_{h_j}\right) - \varepsilon \cdot F(\mathsf{opt}) - \varepsilon L$$

and
$$x_i \leq \mathsf{opt}_i \ . \tag{5}$$

We prove this by induction on $i$. In other words, we prove that the two above inequalities hold for $i \in \{1, 2\}$ given that they holds for every $i' < i$ that belongs to $\{1, 2\}$ (if there is such an $i'$).

The value of $x_i$ is determined by an execution of the algorithm whose existence is guaranteed by Proposition 5.1. Thus, to prove the above inequalities, we need to use the guarantees of this proposition. Moreover, we notice that this is possible since the target value $v_i$ passed to the algorithm of this proposition clearly falls within the allowed range because $\mathsf{opt}_{h_i} \leq u_{h_i}$. Hence, by the first guarantee of Proposition 5.1,

$$F\left(\sum_{j=1}^{i} x_{h_j} \cdot \mathbf{e}_j\right) \geq v_i - \varepsilon L \geq F\left(\sum_{j=1}^{i-1} x_{h_j} \cdot \mathbf{e}_{h_j} + \mathsf{opt}_{h_i}\mathbf{e}_{h_i}\right) - \varepsilon \cdot F(\mathsf{opt}) - \varepsilon L \ .$$

Inequality (4) now follows from the last inequality by subtracting $F\left(\sum_{j=1}^{i-1} x_{h_j} \cdot \mathbf{e}_{h_j}\right)$ from both its sides and observing that, by the submodularity of $F$ and the induction hypothesis,

$$F\left(\sum_{j=1}^{i-1} x_{h_j} \cdot \mathbf{e}_j + \mathsf{opt}_{h_i}\mathbf{e}_{h_i}\right) - F\left(\sum_{j=1}^{i-1} x_{h_j} \cdot \mathbf{e}_j\right) \geq F\left(\sum_{j=1}^{i} \mathsf{opt}_{h_j} \cdot \mathbf{e}_{h_j}\right) - F\left(\sum_{j=1}^{i-1} \mathsf{opt}_{h_j} \cdot \mathbf{e}_{h_j}\right) \ .$$

To prove Inequality (5), we note that the second guarantee of Proposition 5.1 implies that for every $0 \leq y < x_{h_j}$ we have

$$F\left(\sum_{j=1}^{i-1} x_{h_j} \cdot \mathbf{e}_j + y\mathbf{e}_{h_i}\right) < v_i \leq F\left(\sum_{j=1}^{i-1} x_{h_j} \cdot \mathbf{e}_j + \mathsf{opt}_j\mathbf{e}_{h_i}\right) \ ,$$

and therefore, $\mathsf{opt}_{h_j}$ cannot fall in the range $[0, x_{h_j})$. $\qquad\square$

### D.6 Proof of Lemma 5.4

**Lemma 5.4.** *Algorithm 3 outputs a vector $\mathbf{x}_{\mathrm{CA++}}$ whose value is at least $(1 - 1/e - 4\varepsilon) \cdot F(\mathsf{opt}) - \varepsilon(B+2)L$.*

*Proof.* Since $\mathbf{x} \leq \sum_{j=1}^{2} \mathsf{opt}_{h_j} \cdot \mathbf{e}_{h_j}$ by Lemma 5.3, the submodularity of $F$ guarantees that

$$F'\left(\mathsf{opt} - \sum_{j=1}^{2} \mathsf{opt}_{h_j} \cdot \mathbf{e}_{h_j}\right) = F\left(\mathsf{opt} + \sum_{j=1}^{2} (x_{h_j} - \mathsf{opt}_{h_j}) \cdot \mathbf{e}_{h_j}\right) - F(\mathbf{x})$$

$$\geq F(\mathsf{opt}) - F\left(\sum_{j=1}^{2} \mathsf{opt}_{h_j} \cdot \mathbf{e}_{h_j}\right) \ .$$

Therefore, since $\mathsf{opt} - \sum_{j=1}^{2} \mathsf{opt}_{h_j} \cdot \mathbf{e}_{h_j}$ is one feasible solution for the instance received by Algorithm 4.1, we get by Lemma 4.1 that there exists a coordinate $i \in [n] \setminus \{h_1, h_2\}$ such that[4]

$$F'(\mathbf{x}_{\mathrm{CA}}) \geq (1 - 1/e - 2\varepsilon) \cdot F'\left(\mathsf{opt} - \sum_{j=1}^{2} \mathsf{opt}_{h_j} \cdot \mathbf{e}_{h_j} - \mathsf{opt}_i\mathbf{e}_i\right) - \varepsilon BL$$

$$\geq (1 - 1/e - 2\varepsilon) \cdot \left[F'\left(\mathsf{opt} - \sum_{j=1}^{2} \mathsf{opt}_{h_j} \cdot \mathbf{e}_{h_j}\right) - F'(\mathsf{opt}_i\mathbf{e}_i)\right] - \varepsilon BL$$

$$\geq (1 - 1/e - 2\varepsilon) \cdot \left[ F(\mathsf{opt}) - F\left( \sum_{j=1}^{2} \mathsf{opt}_{h_j} \cdot \mathbf{e}_{h_j} \right) - F'(\mathsf{opt}_i \mathbf{e}_i) \right] - \varepsilon BL$$

$$\geq (1 - 1/e - 2\varepsilon) \cdot \left[ F(\mathsf{opt}) - \frac{3}{2} \cdot F\left( \sum_{j=1}^{2} \mathsf{opt}_{h_j} \cdot \mathbf{e}_{h_j} \right) \right] - \varepsilon BL \ ,$$

where the second inequality follows from the submodularity of $F$, and the last inequality holds since the submodularity of $F$ and the definitions of $h_1$ and $h_2$ imply

$$F'(\mathsf{opt}_i \mathbf{e}_i) = F\left( \sum_{j=1}^{2} \mathsf{opt}_{h_j} \cdot \mathbf{e}_{h_j} + \mathsf{opt}_i \mathbf{e}_i \right) - F\left( \sum_{j=1}^{2} \mathsf{opt}_{h_j} \cdot \mathbf{e}_{h_j} \right)$$

$$\leq \frac{1}{2} \left[ F(\mathsf{opt}_i \mathbf{e}_i) - F(\mathbf{0}) + F(\mathsf{opt}_{h_1} + \mathsf{opt}_i \mathbf{e}_i) - F(\mathsf{opt}_{h_1} \mathbf{e}_{h_1}) \right]$$

$$\leq \frac{1}{2} \left[ F(\mathsf{opt}_{h_1} \mathbf{e}_{h_1}) - F(\mathbf{0}) + F(\mathsf{opt}_{h_1} + \mathsf{opt}_{h_2} \mathbf{e}_{h_2}) - F(\mathsf{opt}_{h_1} \mathbf{e}_{h_1}) \right]$$

$$= \frac{1}{2} \left[ F(\mathsf{opt}_{h_1} \mathbf{e}_{h_1} + \mathsf{opt}_{h_2} \mathbf{e}_{h_2}) - F(\mathbf{0}) \right] \leq \frac{F(\mathsf{opt}_{h_1} \mathbf{e}_{h_1} + \mathsf{opt}_{h_2} \mathbf{e}_{h_2})}{2} \ .$$

We are now ready to calculate the value of $\mathbf{x}_{\mathsf{CA++}} = \mathbf{x} + \mathbf{x}_{\mathsf{CA}}$. By the above calculation and Lemma 5.3,

$$F(\mathbf{x} + \mathbf{x}_{\mathsf{CA}}) = F'(\mathbf{x}_{\mathsf{CA}}) + F(\mathbf{x})$$

$$\geq (1 - 1/e - 2\varepsilon) \cdot \left[ F(\mathsf{opt}) - \frac{3}{2} \cdot F\left( \sum_{j=1}^{2} \mathsf{opt}_{h_j} \cdot \mathbf{e}_{h_j} \right) \right] - \varepsilon BL$$

$$+ F\left( \sum_{j=1}^{2} \mathsf{opt}_{h_j} \cdot \mathbf{e}_{h_j} \right) - 2\varepsilon \cdot F(OPT) - 2\varepsilon L$$

$$\geq (1 - 1/e - 4\varepsilon) \cdot F(\mathsf{opt}) - \varepsilon(B + 2)L \ . \qquad \square$$

## D.7  Proof of Observation 5.5

**Observation 5.5.** *Consider the vector $\mathbf{x}$ at the point in which Algorithm 3 guesses the value $v_i$. Then, there exists a value in the set $\mathcal{J}(\mathbf{x}, h_i)$ obeying the requirements from $v_i$.*

*Proof.* Let $j$ be the maximal integer for which $F(\mathbf{x}) + \varepsilon j \cdot F(u_{h_i} \mathbf{e}_{h_i}) \leq F(\mathbf{x} + \mathsf{opt}_{h_i} \mathbf{e}_{h_i}) \leq F(\mathbf{x} + u_{h_i} \mathbf{e}_{h_i})$. Since $F(\mathbf{x}) \leq F(\mathbf{x} + \mathsf{opt}_{h_i} \mathbf{e}_{h_i})$ by the monotonicity of $F$, $j$ is non-negative, and thus, $F(\mathbf{x}) + \varepsilon j \cdot F(u_{h_i} \mathbf{e}_{h_i})$ belongs to $\mathcal{J}(\mathbf{x}, h_i)$ and $F(\mathbf{x}) + \varepsilon j \cdot F(u_{h_i} \mathbf{e}_{h_i}) \geq F(\mathbf{x})$. Furthermore, by the definition of $j$,

$$F(\mathbf{x}) + \varepsilon j \cdot F(u_{h_i} \mathbf{e}_{h_i}) \geq F(\mathbf{x} + \mathsf{opt}_{h_i} \mathbf{e}_{h_i}) - \varepsilon \cdot F(u_{h_i} \mathbf{e}_{h_i}) \geq F(\mathbf{x} + \mathsf{opt}_{h_i} \mathbf{e}_{h_i}) - \varepsilon \cdot F(\mathsf{opt}) \ ,$$

where the second inequality holds since $u_{h_i} \mathbf{e}_{h_i}$ is a feasible solution. Thus, $F(\mathbf{x}) + \varepsilon j \cdot F(u_{h_i} \mathbf{e}_{h_i})$ obeys the requirements from $v_i$. $\qquad \square$

## Footnotes

[4]As stated, Lemma 4.1 applies only to the optimal solution, not to every feasible solution. However, one can verify that its proof does not use the optimality of the solution.