[Reviews · NeurIPS 2020]

Review 1

Summary and Contributions: The paper studies the problem of maximizing a monotone continuous submodular function subject to an l1 constraint. This is a generalization of the problem of maximizing a monotone discrete submodular function subject to a cardinality constraint. The paper shows a near optimal 1-1/e-epsilon approximation. The key technique is to relate this problem to the usual discrete submodular problem with a knapsack constraint.

Strengths: - This is a rare work that generalizes to the continuous submodular function without DR-submodularity. Before this work, there was only a result for box-constrained maximization of non-monotone functions.

Weaknesses: The result seems purely theoretical at the moment without useful models where this generality is required.

Correctness: The claims and proofs seem correct.

Clarity: The paper is reasonably well-written.

Relation to Prior Work: The previous works are clearly discussed. They are missing a reference to Soma and Yoshida but hopefully it will be added to the revision.

Reproducibility: Yes

Additional Feedback: =====After rebuttal====== Please add a discussion on the comparison with Soma-Yoshida. In particular, I think the two settings are very related. The setting in this paper assumes smoothness, which is a very strong assumption compared with bounded gradient, a sufficient condition for approximating the continuous function with a grid. Consider a feasible point x. We will show there is a grid point that is not much worse in objective. Notice that if we subtract epsilon<=B/n^2 from the largest coordinate, wlog assume that is the first coordinate, and add epsilon/n to the rest, it is still feasible. Thus, consider a new point y that is the same as x everywhere except y_1 >= x_1 - epsilon and note that x_1 >= B/n >= n*epsilon. We just need to show that f(y) is not much smaller than f(x) (our grid point even dominates x in coordinates 2,3,.. so its value is even larger than f(y)). Let h be the one dimensional function where h(a) = f(a, x_2, x_3, ..). Let g=h'(y_1). By smoothness and monotonicity, h(y_1)-h(0) >= h(y_1) - h(y_1 - min(y_1, g/L)) >= min(g^2/(2L), y_1*(g-L*y_1/2)) h(x_1)-h(y_1) <= epsilon*g + L(epsilon)^2/2 Case 1. g < y_1*L. We have h(x_1)-h(y_1) <= epsilon*y_1*L + L(epsilon)^2/2 <= L*epsilon*x_1. Case 2. g >= y_1*L. We have h(x_1)-h(y_1) <= 2*epsilon/y_1 * (h(y_1)-h(0)) + L*epsilon^2/2 <= 2*(h(y_1)-h(0))/(n-1) + L*epsilon*x_1/n.


Review 2

Summary and Contributions: This paper studies an optimization problem in which the goal is to maximize a continuous submodular (not necessarily DR-submodular) function under a simple linear constraint. The authors have proposed two algorithms, namely COORDINATE-ASCENT+ and COORDINATE-ASCENT++, to solve this problem and in particular, the latter obtains the tight (1-1/e-\epsilon) approximation ratio. -------------- Update: I read the authors' feedback, thanks for your careful and detailed responses. In particular, I am convinced about the applications of (non-DR) continuous submodular functions and hence the importance of the algorithms provided in this paper. I've increased my score for the paper.

Strengths: This work is significant in several respects. First of all, the continuous submodular maximization problem has not been studied before in the literature. To be precise, continuous DR-submodular maximization, where the objective function is coordinate-wise concave along with continuous submodular, has been extensively studied before, however, the study of continuous submodular (not necessarily DR-submodular) maximization in this paper is novel and could be of high interest to the NeurIPS community. Moreover, although the computational complexity of the proposed algorithm is rather high, the algorithm is parallelizable which makes it useful in practice. The work contains a significant amount of novel proof ideas (mostly provided in the appendix) and overall, the theoretical contributions of the paper are significant.

Weaknesses: Despite all the theoretical strengths of this work, the paper has the following weaknesses: - The main text of the paper is very technical and it is hard for someone with limited exposure to the previous literature on this topic to follow the paper. For instance, it would have been useful to provide the diminishing returns type definition of continuous submodularity and compare it to that of DR-submodular functions. Moreover, a brief overview of techniques for continuous DR-submodular maximization and the reasons why they fail to apply to continuous submodular maximization would have been very beneficial. - The authors have failed to provide set of examples of continuous submodular functions that are not DR-submodular and are used in practice to motivate the study of maximizing such functions. I realize that the provided example $\phi_{i,j}(x_i-x_j)$, where $\phi_{i,j}$ is convex, is indeed continuous (non-DR)-submodular, but in my view, this point needed to be emphasized in the paper. - This paper considers a very simple linear constraint and the authors have failed to mention why their proposed algorithms could not be used for more general convex constraints and what the challenges are to design algorithms for such constraints. - This paper does not contain any numerical experiments to verify the efficiency of their proposed algorithms. In particular, the COORDINATE-ASCENT++ algorithm has a rather high computational complexity and it would have been useful to implement the algorithm and show the claim "the algorithm is easily parallelizable" in practice.

Correctness: All the theoretical claims and results of this paper have been well justified through mathematically sound arguments and proofs. I took a quick look at the proofs in the appendix as well and they looked fine to me. The paper does not contain any numerical experiments.

Clarity: As mentioned earlier, the main text of the paper is a bit too technical and it's hard to follow for the general audience with limited familiarity with the topic. I'm well familiar with the literature and nonetheless, occasionally, I had difficulty verifying the arguments and proof steps. To remedy this issue, I think Algorithm 5 and Algorithm 6, provided in the appendix, need to move to the main text of the paper and be discussed in more detail to help the reader better understand the COORDINATE-ASCENT+ and COORDINATE-ASCENT++ algorithms.

Relation to Prior Work: The authors have done a great job reviewing the previous literature on continuous DR-submodular maximization and have differentiated their contributions to the more general problem of continuous (non-DR)submodular maximization.

Reproducibility: Yes

Additional Feedback: - Discuss the challenges of solving continuous submodular maximization problem subject to general convex constraints. - Provide a list of continuous (non-DR)-submodular functions that are used in practice in different domains to further emphasize the significance of the framework of the paper. - Provide more details and explanations for the steps of the proofs to make it easier to follow.


Review 3

Summary and Contributions: This paper studies maximization of a monotone continuous submodular function under the cardinality constraint. The technical contributions are summarized as - ((e-1)/(2e-1)-ε)-approximation algorithm which performs O(n/ε) iterations with O(n√B/ε + n log n) cost per iteration - (1-1/e-ε)-approximation algorithm which performs O(n/ε) iterations with O(n^3√B/ε^2.5 + n^3 log n/ε^2) cost per iteration. This algorithm can be parallelized to reduce the cost to O(n√(B/ε) + n log n) per iteration, where ε is a user constant and n is the dimension. Previous work focused on a subclass of continuous submodular functions, i.e., DR-submodular functions, which assumes the natural diminishing return property. In contrast, continuous submodular functions satisfy only a weaker condition.

Strengths: - First approximation algorithms for continuous submodular maximization under the cardinality constraint - Approximation ratio is tight

Weaknesses: - A similar work exists in the literature whose time complexity is better (Update: the previous work has a flaw)

Correctness: I found no mathematical error.

Clarity: Clearly written.

Relation to Prior Work: They missed a relevant reference of lattice submodular maximization under cardinality constraint.

Reproducibility: Yes

Additional Feedback: The results presented in this paper look reasonable. However, I found a very similar result in the literature (Soma and Yoshida 2018, Mathematical Programming). They studied maximization of monotone lattice submodular function (hence it corresponds to continuous submodular maximization in this paper) subject to a cardinality constraint. Taking a fine enough discretization of each axis of the domain, one can reduce the continuous setting to the lattice setting. Soma-Yoshida's algorithm outputs a (1-1/e-ε)-approximate solution in O(n/ε^2 log ||u||_∞ log B/ε log τ) time, where τ is the minimum positive increase of the objective function. Therefore, it is faster than the proposed algorithms above. Indeed, Algorithm 1 in this NeurIPS manuscript looks very similar to Soma-Yoshida's Algorithm 3: Both algorithms are greedy with approximate step size search for each axis direction. Furthermore, their subroutine (Proposition 3.1) indeed does discrete step size search for each axis, so these two algorithms are similar even conceptually. Of course, the continuous submodular setting has several benefits over the integer lattice setting such as wider applications in ML and more natural integration with continuous optimization (Frank-Wolfe, SGD, step size, etc). But this paper focuses on the purely algorithmic side and the algorithms itself are based on discretization, so the advantage over the integer lattice is quite unclear. Unfortunately, the authors seem to have missed the Soma-Yoshida paper and did not give a detailed comparison. I would like to see differences (if there is any) in the rebuttal. ### update after rebuttal ### I have read the authors' feedback and agree that the previous result in the reference I pointed out were not completely true. The algorithm needs to be modified with partial enumeration, which is also used in this submission. This makes the complexity of the previous lattice setting algorithm to something at least n^3, which addresses my first comment. Also, it was nice the authors addressed my second comment by explaining the difference between their setting and the integer lattice setting. I hope the authors include them in the revised version. I raised my score.


Review 4

Summary and Contributions: This paper proposes a variant of the coordinate ascent algorithm for monotone continuous submodular maximization with a linear constraint. The proposed algorithm achieves (1-1/e-epsilon)-approximation without DR-submodularity.

Strengths: This paper deals with the problem of maximizing F(x) subject to ||x||_1 <= B, where F is a monotone L-smooth continuous submodular (not necessarily DR-submodular) function. Since the coordinate-wise concavity does not hold for continuous submodular functions, it is hard to apply existing methods such as continuous greedy to this problem. The authors propose two variants of the coordinate ascent, which achieve ((e-1)/(2e-1)-eps)-approximation in O(n/epsilon) time and (1-1/e-eps)-approximation in O(n^3/epsilon^2.5 + n^3 log(n)/epsilon^2) time, respectively. To my knowledge, the idea of applying the coordinate ascent to continuous submodular maximization is novel.

Weaknesses: In terms of applications, continuous submodular maximization is not highly motivated. The problem this paper tackles is continuous submodular maximization, which is a generalization of DR-submodular maximization. There are already efficient and effective algorithms for DR-submodular maximization, and continuous submodular functions that appear in most applications satisfy DR-submodularity. Therefore it is hard to say the proposed algorithm is ready to be applied to real world problems immediately. Also, this paper contains small typos, though they are not critical.

Correctness: I skimmed all the proofs and found no error.

Clarity: The overall writing quality is good.

Relation to Prior Work: This paper sufficiently surveys existing work on continuous submodular maximization. I think it would be better to mention the following work by Soma and Yoshida. Maximizing monotone submodular functions over the integer lattice. Mathematical Programming (2018). In this Soma--Yoshida paper, they provide a (1 - 1/e)-approximation algorithm for monotone submodular (not DR-submodular) maximization on the integer lattice with a cardinality constraint. This problem can be regarded as a discrete analogue of the problem tackled in this paper.

Reproducibility: Yes

Additional Feedback: - p.4, l.141--143: It would be helpful if the authors mention the case when x_i > y_i. In this case, x_i can never be equal to y_i, which deviate from the explanation of ``good'', but all analyses in the proof hold. - p.5, l.162: ] could be removed. - p.6, l.197: ||_1 could be removed. - p.6, l.198 and l.205: |_1 could be added. - p.6, l.201--206: \ell might be replaced with \ell'. - p.7, l.247: OPT could be replaced with opt. - p.17, l.531--532: B could be replaced with B^0.5 in three places. # Update after the author feedback My main concern is about the applications of (non-DR) continuous submodular functions. In the feedback, the authors mentioned the problem of finding the mode of MTP2 distributions. I think this is a reasonable application and agree with the acceptance of this paper.

[Author Response · NeurIPS 2020]

We would like to thank all the reviewers for their thoughtful remarks. We will revise the manuscript to fix the typos
pointed out by Reviewer 4 and implement the suggestions of Reviewer 2 regarding ways to improve the readability of
the paper. In the following, we address the main concerns raised by the reviewers.

**Q1:** Applications and examples of continuous submodular functions beyond DR-submodularity.

**A1:** Maybe the simplest example of a continuous submodular function that is not DR-submodular is the quadratic
function $F(x) = x^T H x + h^T x + b$ where only the off-diagonal entries of $H$ are non-positive (and there is no restrictions
on the diagonal entries). Moreover, as we mentioned in the introduction, continuous submodular functions naturally
arise as the negative log-densities of probability distributions. For instance, a distribution $p$ on $\mathcal{X}$ is called multivariate
totally positive of order 2 (MTP2) if $p(x)p(y) \leq p(x \vee y)p(x \wedge y)$ for all $x, y \in \mathcal{X} \subset \mathbb{R}$. MTP2 implies positive
association between random variables. As an example, a multivariat Gaussian distibution is MTP2 if an only if its
inverse covariance matrix has non-positive off-diagonal entries. Therefore, finding the the most likely configuration in
this setting amounts to maximizing a continuous submodular function. Finally, as we mentioned in the paper, finding
the mode of multivariate logistic, Gamma and F distributions, as well as characteristic roots of random Wishart matrices
amounts to maximizing a continuous submodular function.

**Q2:** Why the proposed method cannot handle more complicated constrains?

**A2:** As mentioned in the paper, our algorithms resemble algorithms for maximizing a submodular *set* function subject
to a knapsack constraint. Moreover, this is the case even when all the coordinates have a coefficient of 1 in the constraint,
which corresponds to a simpler cardinality constraint. An intuitive reason for that is that we view a monotone non-DR
submodular continuous function as a monotone submodular *set* function over an infinite ground set consisting of pairs
$(i, v)$, where $i$ is a coordinate and $v$ is the value assigned to this coordinate. Unfortunately, this intuitive reduction
converts cardinality constraints into knapsack constraints, forcing us to employ knapsack techniques even when handling
cardinality-like constraints. Similarly, other constraints will also become more involved if the above intuitive reduction
is applied to them, usually leading to constraints that are difficult to handle.

**Q3:** Relationsip between the current work and the paper by Soma and Yoshida.

**A3:** We would like to thank Reviewers 3 and 4 for pointing out to us the paper of Soma and Yoshida, which is indeed
related to ours. However, despite this relationship, it is not easy to derive a result for our setting based on the paper Soma
and Yoshida for a few reasons. Perhaps the most significant of these reasons is that we believe (and they confirmed)
that the result of Soma and Yoshida is not entirely correct, as is hinted by the lack of an enumeration step in their
algorithm. Specifically, the main issue is that the proof of their Lemma 5 invokes Lemma 4 with $k' + \Delta(a)$, but without
verifying that $k' + \Delta(a)$ is upper bounded by $k_{\max}$ (which is a necessary condition of Lemma 4). We have indeed
contacted Soma and Yoshida regarding this error. Here is a part of their response: "Yuichi and I (Tasuku) discussed the
issue and confirmed that our algorithm has a flaw. As you pointed out, Lemma 4 cannot be applied especially when
$k_{\max} = r - y(E)$. We did not find out an easy fix".

**Q4:** Can the work by Soma and Yoshida be easily extended to the continuous setting?

**A4:** Even if the result of Soma and Yoshida was corrected, extending it to our setting is non-trivial. There are two
natural ways in which one might try to achieve this goal. One way is to try to create a "black box" reduction from
our continuous setting to their discrete setting by considering a fine enough lattice. This could only work if the first
derivative of the objective function was bounded. However, when the first derivative can be very large (which is allowed
in our setting), there might be a significant difference between the maximum objective value achievable at the *feasible*
points of the lattice and the real maximum objective value. The other way in which one might try to extend the result
of Soma and Yoshida to our setting is by employing ideas from this result in the creation of a new algorithm for our
continuous setting. As explained in the reviews, some of our ideas overlap with those of Soma and Yoshida. However,
our work deviates significantly from that of Soma and Yoshida, and cannot be viewed as an easy derivative of their
work. Here are two pieces of evidence for that.
1) The time complexity of Soma and Yoshida depends on $\tau$—the ratio between the maximum value of a solution
whose support consists of a single coordinate and the minimum increase in the value of a solution when its $\ell_1$ norm is
increased by 1. This definition of $\tau$ is strongly connected to the discrete nature of the problem considered by Soma and
Yoshida as it represents the ratio between the maximum and minimum non-zero contributions that a single coordinate
can have. Extending $\tau$'s definition to our continuous domain is problematic since continuity means that a coordinate
can contribute an arbitrarily low amount, leading to an infinite value for $\tau$. Instead, we force our algorithm to increase
the $\ell_1$ norm of its solution by at least some minimal amount in each iteration, which requires us to develop a bound on
the loss due to this extra restriction.
2) The algorithm of Soma and Yoshida heavily depends on binary searches, which are natural in their discrete setting,
but make little sense in our continuous setting. This required us to develop the more sophisticated techniques represented
by Propositions 3.1 and 5.1.

[Meta-Review · NeurIPS 2020]

The paper was discussed among the reviewers after the rebuttal phase. Most reviewers found the rebuttal helpful and clarifying, and updated their scores and reviews. The consensus is to accept the paper as a poster in NeurIPS. A reviewer commented that while "the setting in this work is restricted and specialized, overall, the paper provides a comprehensive study of this setting, and after some minor edits to improve the readability, it is in a good shape to be accepted." One concern was "the importance of studying (non-DR) continuous submodular functions in practice" which the authors' rebuttal helped address. The authors should discuss these motivating applications (from the rebuttal) in the paper itself, and attempt to include even more examples and applications to strengthen the motivation for the study of this specific class of problems. Another concern of reviewers was that an exiting paper (Soma-Yoshida 2018) gives a better result, however, the authors' rebuttal pointed out a flaw in the existing paper. In the discussions following the rebuttal, the reviewers found this point valid. Additional comment: the 1/2-approximation guarantee for monotone DR-submodular maximization via gradient descent is likely older than the 2017 paper cited in Related Work (possibly in older work by Chekuri et al.); the authors should look into this and update if needed.